

# Pleistocene reefs of the Egyptian Red Sea: environmental change and community persistence

Lorraine R. Casazza

School of Science and Engineering, Al Akhawayn University, Ifrane, Morocco

## ABSTRACT

The fossil record of Red Sea fringing reefs provides an opportunity to study the history of coral-reef survival and recovery in the context of extreme environmental change. The Middle Pleistocene, the Late Pleistocene, and modern reefs represent three periods of reef growth separated by glacial low stands during which conditions became difficult for symbiotic reef fauna. Coral diversity and paleoenvironments of eight Middle and Late Pleistocene fossil terraces are described and characterized here. Pleistocene reef zones closely resemble reef zones of the modern Red Sea. All but one species identified from Middle and Late Pleistocene outcrops are also found on modern Red Sea reefs despite the possible extinction of most coral over two-thirds of the Red Sea basin during glacial low stands. Refugia in the Gulf of Aqaba and southern Red Sea may have allowed for the persistence of coral communities across glaciation events. Stability of coral communities across these extreme climate events indicates that even small populations of survivors can repopulate large areas given appropriate water conditions and time.

## INTRODUCTION

Coral reefs worldwide are threatened by habitat degradation due to coastal development, pollution run-off from land, destructive fishing practices, and rising ocean temperature and acidification resulting from anthropogenic climate change (*Wilkinson, 2008*; *Poloczanska et al., 2013*; *Pandolfi, 2015*). With increasing concern about the future of coral-reef ecosystems has come increased efforts to predict their fate under varying climate predictions (see review by *Donner, Heron & Skirving, 2009*). Modern ecological studies of reefs seeking to predict future response are limited to analyses at the scale of decades, which may not be indicative of ecological trends at longer time scales (*Denny et al., 2004*; *Pandolfi, 2011*). The fossil record provides unique opportunities to study diversity on longer time scales, and under different environmental conditions (*Jackson & Erwin, 2006*; *Pandolfi, 2011*), which makes it a valuable resource for understanding how coral reefs respond to changing climate regimes.

The Red Sea provides a unique natural laboratory to study the history of coral survival and recovery in the context of environmental catastrophe. Fossil coral terraces from

Corresponding author
Lorraine R. Casazza,
lorraine.casazza@gmail.com

interglacial periods of the Middle and Late Pleistocene occur alongside modern day fringing reefs all along the Red Sea coast (*Plaziat et al., 2008*). These preserved stages of reef growth are punctuated by periods when vast areas of the Red Sea experienced hypersaline conditions unsuitable for most life (*Almogi-Labin, 1982*; *Almogi-Labin, Hemleben & Meischner, 1998*; *Badawi, Schmiedl & Hemleben, 2005*; *Fenton et al., 2000*; *Hemleben et al., 1996*; *Thunnell, Locke & Williams, 1988*). There is ongoing debate about whether reef taxa survived glacial periods in refugia located in the Gulf of Aqaba and/or the southern Red Sea, or if they recolonized the Red Sea from the Gulf of Aden during each interglacial period (*DiBattista et al., 2015*). However, there is general agreement that reefs of the Egyptian coast were most likely devastated by glacial conditions (*Gvirtzman et al., 1977*; *Sheppard & Sheppard, 1991*; *Taviani, 1998*; *Coles, 2003*; *DiBattista et al., 2015*).

This study is an ecological survey of Middle and Late Pleistocene coral assemblages from the Egyptian Red Sea coast, and places them in the wider context of changing Quaternary climate. The aim is to (1) characterize the diversity and paleoenvironments of these coral communities, (2) compare coral taxa between the Middle Pleistocene and Late Pleistocene, and (3) compare taxa between Pleistocene and Modern reefs, in order to determine if and how communities may have changed across glaciation events.

## Geologic setting

The Red Sea is a long, narrow, marginal sea at the western-most extent of the Indo-Pacific (see Fig. 1). It stretches over 2,000 km from the Gulf of Suez in the north to the Straits of Bab el Mandeb in the south. The basin is almost completely enclosed by land, bordered to the east by the Arabian Peninsula, the African continent to the west, and Sinai Peninsula to the north. It is connected to the Indian Ocean via the Gulf of Aden through the straits of Bab el Mandeb, which at just 20 km across and 137 m deep at the Hanish Sill (*Werner & Lange, 1975*), limits water mass exchange. This limited exchange combined with a lack of fresh water input from rivers and the surrounding hot and arid climate accounts for an average surface salinity of 40–41‰. It is an active, maritime rift system overlying the divergent plate boundary between the African and Arabian plates, and connecting the East African Rift Valley to the southwest, to the Dead Sea rift to the northeast.

During interglacial high stands of the Pleistocene (as well as the present) extensive fringing reefs developed along the coasts of the Red Sea. Today, emerged reef terraces running parallel to the modern coastline are a nearly continuous feature of the entire Red Sea (*El Moursi et al., 1994*; *Gvirtzman et al., 1977*; *Plaziat et al., 2008*; *Plaziat et al., 1998*). On the Egyptian coast, Late Pleistocene terraces form a low cliff along the water line. In most locations they appear as two obvious terraces, the lower at approximately 1.5 m above mean sea level and the upper terrace approximately 4 m above mean sea level. If the Egyptian coast south of the Gulf of Suez has been tectonically stable over the Late Pleistocene (*Hoang & Taviani, 1991*; *Bosworth & Taviani, 1996*; *Plaziat et al., 2008*), then their current elevation is close to their original elevation. However, *Lambeck et al. (2011)* have proposed that the area has undergone long-term

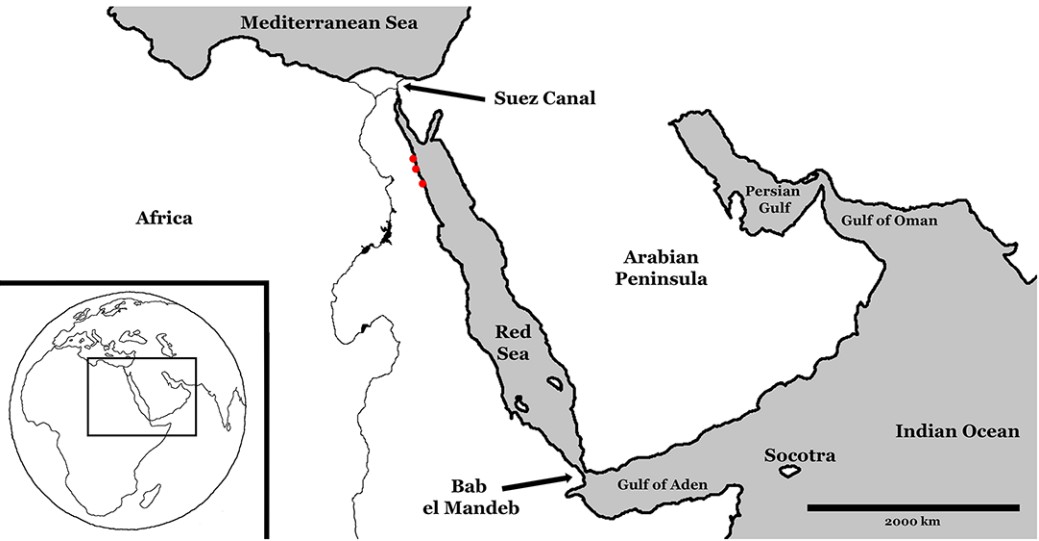

**Figure 1** **Modern Red Sea region.** Red dots represent study sites, from north to south: Sharm Al Arab, Wadi Gawasis, and Wadi Wizr. During Pleistocene glaciation events, sea level fell and exchange with the Indian Ocean was restricted.

tectonic uplift. In this scenario the reef terraces could be several meters higher than they were at the time of reef formation (*Lambeck et al., 2011*).

Although *El Moursi et al. (1994)* interpreted these Late Pleistocene terraces as three independent stages of reef growth, an exhaustive review of uranium series dates by *Plaziat et al. (2008)* indicated both Late Pleistocene terraces belong to MIS 5e, with most ages around 123,000 years before present (bp), and suggested that the platform morphology is a result of erosion. Wadis (erosional valleys) running perpendicular to the coast form breaks in the terraces and modern fringing reefs, allowing access to the outcrop face.

Middle Pleistocene reefs occur farther inland and at elevations up to 50 m as a result of sea-level change and tectonic uplift (*El-Asmar, 1997*; *El Moursi et al., 1994*; *Gvirtzman et al., 1977*; *Plaziat et al., 1998*, *2008*). These have been attributed to MIS 7 and 9, and a limited number of uranium series dates have placed them at around 200,000 years bp (MIS 7), with some ages over 300,000 years bp (MIS 9) (*El-Asmar, 1997*; *El Moursi et al., 1994*; *Gvirtzman, 1994*; *Hoang & Taviani, 1991*; *Plaziat et al., 2008*).

Based on elevation and preservation, this study may include terraces from MIS 5, MIS 7, and MIS 9. However, given the uncertainty of ages on the older terraces, they are simply referred to the Middle Pleistocene and the younger terraces are Late Pleistocene. The underlying assumption is that the terraces studied here are separated by a single glacial period (MIS 6) with the understanding that it may be an underestimation of the spanned time period. This is a more conservative assumption than the alternative because dates over 300,000 years bp are even less certain than those for MIS 7.

## Paleontology of Red Sea reefs

The Pleistocene reef terraces of the Red Sea coast have attracted geological study for their contribution to past sea-level reconstructions (*Gvirtzman, 1994*; *Hemleben et al., 1996*;

*Siddall et al., 2004*; *Thunnell, Locke & Williams, 1988*), but much less work has focused on the biology of the reef fauna. *Dullo (1990)* described Late Pleistocene fauna from the terraces of Saudi Arabia, and *El-Sorogy (1997*, *2002*, *2008)* described the corals of Middle and Late Pleistocene fauna from the Sinai Peninsula, and Pleistocene of the Egyptian coast. *Al-Rifaiy & Cherif (1988)* provide a more limited description of corals from the coast of Jordan, and likewise *Bruggemann et al. (2004)* mention coral species in descriptions of terraces from the coast of Eritrea. More recently *Alexandroff, Zuschin & Kroh (2016)* provided a quantitative comparison of Scleractinian genera between Late Pleistocene and Modern reef habitats on the Egyptian coast. The local geography and political climate of the region limit access to outcrops, so these works provide a valuable resource for comparing the fossil fauna of the Red Sea to the modern fauna.

## METHODS

### Study area

I selected field sites based on three criteria: (1) the presence of coral-reef terraces representing both Late and Middle Pleistocene interglacials, (2) at least 100 m of exposed outcrop for each terrace, and (3) safe and legal access to the outcrops. Three locations fulfilled these criteria: Sharm Al Arab (26°57′58″N, 33°54′40″E), Wadi Gawasis (26°33′18″N, 34°02′06″E), and Wadi Wizr (25°47′08″N, 34°29′05″E) (see Fig. 2). All necessary permits were obtained for the described study, which complied with all relevant regulations. The permit was issued by the Egyptian Environmental Assessment Agency (EEAA) and authorized by Dr. Moustafa Fouda. All fossil material was reviewed and approved for export by the Egyptian Geologic Museum in Cairo.

### Quantitative methods

Line-point intercept transects of at least 100 m length (and up to 135 m length) were used to identify coral species on all the examined terraces. A 30 m tape measure was laid along the terrace at a constant height oriented parallel to the ancient shoreline, and photographs were taken with an Olympus Stylus 740 digital camera every meter to record identifying corallite details and the colony morphologies of reef-building coral species (the solitary corals of family Fungiidae were not included). In some cases, multiple species intersected a meter mark on the transect, and in these cases all species were identified and photographed. Sediment types were also identified when they intersected a meter mark on the transect. They were characterized based on mineral type, grain size, and degree of sorting, which were visually estimated in the field using a hand lens, ruler, and grain size chart. Seven of the eight terrace transects contain gaps in transect data due to covered, absent or inaccessible stretches of the outcrop. I did transects on both the lower (1.5 m) and upper (4 m) platforms of the Late Pleistocene terraces at Wadi Gawasis and Wadi Wizr, but only the lower (1.5) platform at Sharm Al Arab. The Middle Pleistocene terraces studied at Sharm Al Arab and Wadi Wizr occur at approximately 6 m and the Middle Pleistocene terrace at Wadi Gawasis occurs at around 12 m.

I used the transect data to calculate estimates of percent coral cover, species abundances, Shannon–Wiener index, and Simpson's diversity index. Rarefaction and

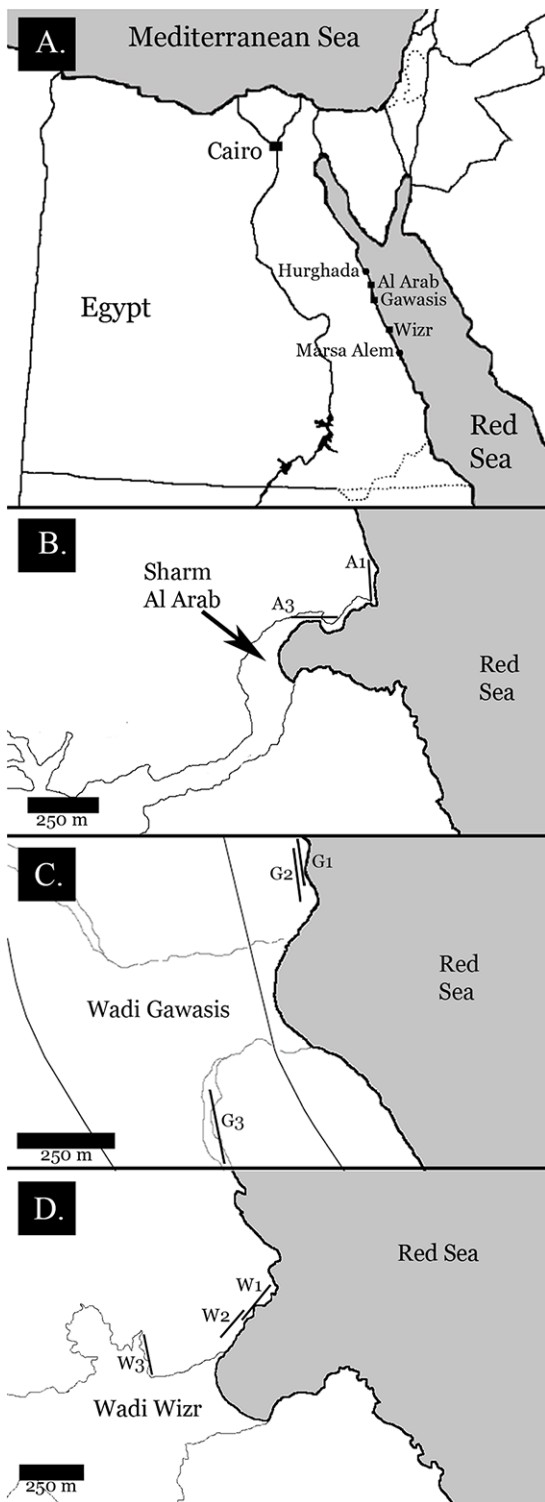

**Figure 2 Field location maps.** (A) Egyptian coastline, showing field sites between Hurghada and Marsa Alem; (B) Sharm Al Arab field site showing the location of fossil reef terraces. A1 is the lower Late Pleistocene terrace, A3 is the Middle Pleistocene terrace; (C) Wadi Gawasis field site showing the location of fossil reef terraces. G1 is the ower Late Pleistocene terrace, G2 is the upper Late Pleistocene terrace, G3 is the Middle Pleistocene terrace; (D) Wadi Wizr field site showing the location of fossil reef terraces. W1 is the lower Late Pleistocene terrace, W2 is the upper Late Pleistocene terrace, W3 is the Middle Pleistocene terrace.

rank abundance curves were also generated to compare diversity between terraces. Differences between coral assemblages were calculated using the abundance-based Chao dissimilarity index, which is robust even when comparing samples of different sizes (*Chao et al., 2005*). All analyses were performed in R 3.3.0 (*R Core Team, 2016*) using the vegan package (*Oksanen et al., 2016*). For the rarefaction curves I sourced code authored by Joshua Jacob (http://www.jennajacobs.org/R/rarefaction.txt).

## Qualitative methods

Corals were identified to the species level using *Ditlev (1980)*, *Scheer & Pillai (1983)*, *Sheppard & Sheppard (1991)*, and *Veron (2000)* and photographic plates from W.-C. Dullo, 1990, unpublished data. In the case of very weathered specimens from the Middle Pleistocene terraces, corals were embedded in epoxy under vacuum, and longitudinal and latitudinal thin sections were prepared from the better preserved interiors to aid in identification. When coral preservation was too poor to identify species, specimens were identified to the generic or family level, depending on what diagnostic features were still observable. Taxonomic names were checked in the World Register of Marine Species (WoRMS) and updated to reflect current taxonomic treatments (see Table 1 legend for specific citations). Specimens of the most commonly occurring and difficult-to-identify reef species were collected and deposited at the University of California Museum of Paleontology (coral specimen numbers 557184–557253, other invertebrate specimen numbers 557273–557391; see Table S1 for details).

The reef environment (reef zone) preserved on each fossil terrace was determined by comparing the relative abundance of coral species, coralline algae, and sediments (*Perrin, Bosence & Rosen, 1995*) with those recorded from coral communities of the modern Red Sea (*Loya, 1972*; *Sheppard & Sheppard, 1991*; *Riegl & Velimirov, 1994*; *Riegl & Pillar, 1999*). Taxa found on the Pleistocene terraces are also compared with a modern regional taxonomic dataset published by *DiBattista et al. (2015)* to determine if any taxa may have gone regionally extinct.

## RESULTS

A total of 1,141 coral or sediment identifications were collected: 719 from the Late Pleistocene and 422 from the Middle Pleistocene. Of these, 14 were unusable because coral specimens were too eroded or damaged to identify to the family level, and so they were removed from the data set. Sediments and coralline algae accounted for 195 data points, and 932 were coral taxa.

### Sediments

Only four distinct sediment types were identified: coralline algae, marine sand, non-marine sand, and carbonate mud. The coralline algae was recognizable by its distinctive growth forms: sometimes rhodolithic and sometimes encrusting (see Fig. 3A). Marine sand was primarily composed of poorly sorted angular and sub-angular fragments of shell, coral, calcifying algae, and urchin spines (see Fig. 3B), with grains ranging from sand to gravel size. This is consistent with modern reef sediments, which are

**Table 1 Reported occurrences of Pleistocene Scleractinian coral in the Red Sea basin from the Middle Pleistocene to present.**

| Pleistocene corals | Middle Pleistocdene | Late Pleistocene | Modern |
|---|---|---|---|
| *Acanthastrea echinata*[1,2,5] | ? | ■ | ■ |
| *Acanthastrea rotundoflora*[1] | | ■ | |
| *Acropora clathrata*[5] | ? | ? | ■ |
| *Acropora cytherea*[5] | ? | ? | ■ |
| *Acropora forskali*[1] | ? | ? | |
| *Acropora haimei*[2] | | ■ | |
| *Acropora hemprichii*[5] | ? | ? | ■ |
| *Acropora humilis*[2] | | ■ | |
| *Acropora hyacinthus*[2,5] | ? | ? | ■ |
| *Acropora intermedia*[5] | ? | ? | |
| *Acropora latistella*[5] | ? | ? | |
| *Acropora pharaonis*[2,5] | ? | ? | |
| *Acropora robusta*[5] | ? | ? | |
| *Acropora abrotanoides*[5] | ? | ? | |
| *Acropora valida*[5] | ? | ? | ■ |
| *Astrea curta*[1,5] | ■ | ? | ■ |
| *Astreopora myriophthalma*[2] | | ■ | |
| *Blastomussa merleti*[5] | ? | ? | |
| *Cantharellus doederleini*[5] | ? | ? | |
| *Caulastraea tumida*[1] | | ■ | |
| *Coelastrea aspera*[5] | ? | ? | |
| *Coscinaraea columna*[5] | ? | ? | ■ |
| *Coscinaraea monile*[2,3,5] | ? | ■ | |
| *Ctenactis echinata*[2,5] | ? | ? | |
| *Cynarina lacrymalis*[2] | | ■ | |
| *Cycloseris costulata*[2] | | ■ | |
| *Cycloseris cyclolites*[3,5] | ? | ■ | ■ |
| *Cycloseris tenuis*[5] | ? | ? | |
| *Cycloseris patelliformia*[5] | ? | ? | |
| *Cycloseris vaughani*[2] | | ■ | ■ |
| *Cyphastrea microphthalma*[1,2,5] | ■ | ■ | ■ |
| *Cyphastrea serailia*[1,2,5] | ■ | | |
| *Dipsastraea laxa*[2,5] | ? | ? | |
| *Dipsastraea danai*[5] | ? | ? | |
| *Dipsastraea favus*[1,2,5] | ? | ■ | |
| *Dipsastraea helianthoides*[5] | ? | ? | |
| *Dipsastraea lacuna*[5] | ? | ? | |
| *Dipsastraea lizardensis*[5] | ? | ? | ■ |

(Continued)

| Pleistocene corals | Middle Pleistocene | Late Pleistocene | Modern |
|---|---|---|---|
| *Dipsastraea matthaii[1] | | ■ | ■ |
| *Dipsastraea pallida[1,2,5] | ■ | ■ | ■ |
| *Dipsastraea rotumana[5] | ? | ? | |
| *Dipsastraea speciosa[1,2,4,5] | ■ | ■ | ■ |
| *Dipsastraea veroni[5] | ? | ? | |
| Echinophyllia aspera[2] | | ■ | ■ |
| Echinopora gemmacea[1,2,3,5] | ? | ■ | ■ |
| Echinopora lamellosa[1,2,4,5] | ■ | ■ | ■ |
| Erythrastrea flabellata[1] | | ■ | ■ |
| Favites abdita[1,2,4,5] | ? | ■ | ■ |
| Favites acuticollis[2,5] | ? | ? | ■ |
| Favites chinensis[3] | | ■ | ■ |
| Favites complanata[2,5] | ? | ? | ■ |
| Favites flexuosa[1,2,5] | ? | ■ | ■ |
| Favites halicora[5] | ? | ? | ■ |
| Favites micropentagonus[1] | ■ | | |
| Favites paraflexuosus[1] | | ■ | ■ |
| Favites pentagona[1,2,3,5] | ? | ■ | ■ |
| *Favites rotundata[2] | | ■ | ■ |
| Favites vasta[1] | | ■ | ■ |
| Galaxea astreata[2,5] | ? | ? | ■ |
| Galaxea fascicularis[1,2,3,5] | ■ | ■ | ■ |
| Gardineroseris planulata[1,2] | | ■ | ■ |
| Goniastrea pectinata[1,2,3,5] | ? | ■ | ■ |
| Goniastrea retiformis[1,2,3,5] | ■ | ■ | ■ |
| *Goniastrea stelligera[1,2,5] | | ■ | ■ |
| Goniopora columna[3] | | ■ | ■ |
| *Goniopora pedunculata[5] | ? | ? | |
| Goniopora planulata[2,5] | ? | ■ | ■ |
| Goniopora savignyi[2,3] | | ■ | ■ |
| Goniopora tenella[2] | | ■ | ■ |
| Gyrosmilia interrupta[1] | ■ | | |
| Hydnophora exesa[5] | ? | ? | |
| Hydnophora microconos[1,2,3,5] | ? | ■ | ■ |
| Leptastrea bottae[1,3,4,5] | ■ | ■ | ■ |
| Leptastrea pruinosa[1] | | ■ | ■ |
| Leptastrea purpurea[2] | | ■ | ■ |
| Leptastrea transversa[3,5] | ? | ? | ■ |
| Leptoria phrygia[1,2,5] | ■ | ■ | ■ |

| Table 1 (continued). | | | |
|---|---|---|---|
| **Pleistocene corals** | **Middle Pleistocdene** | **Late Pleistocene** | **Modern** |
| *Leptoseris yabei[2] | | ● | ● |
| Lobophyllia corymbosa[2,3,5] | ? | ● | ● |
| *Lobophyllia erythraea[2] | | ● | ● |
| Lobophyllia hemprichii[1,2,5] | ? | ● | ● |
| Montipora spongiosa[5] | ? | ? | ● |
| Mycedium elephantotus[2] | | ● | ● |
| *Oulophyllia bennettae[3] | | ● | ● |
| Pachyseris speciosa[2] | | ● | ● |
| *Paragoniastrea australensis[5] | ? | ? | ● |
| *Paramontastraea peresi[2,5] | ? | ? | ● |
| Pavona bipartita[1] | ● | | ● |
| Pavona cactus[1,2] | | ● | ● |
| Pavona decussata[1] | ● | | ● |
| Pavona explanulata[2] | | ● | ● |
| Pavona frondifera[1] | | ● | ● |
| Pavona maldivensis[1] | | ● | ● |
| Pavona minuta[1,2] | | | ● |
| Pavona venosa[1] | | ● | ● |
| Platygyra acuta[1] | | ● | ● |
| Platygyra crosslandi[3,5] | ? | ? | ● |
| Platygyra daedalea[1,2,4,5] | ? | ? | ● |
| Platygyra lamellina[1,3] | | ● | ● |
| Platygyra sinensis[1,5] | ● | ● | ● |
| Plerogyra sinuosa[2] | | ● | ● |
| Plesiastrea versipora[5] | ? | ? | ● |
| Pocillopora damicornis[1,2,5] | ● | ● | ● |
| Pocillopora verrucosa[2,5] | ? | ● | ● |
| Porites rus[2,5] | ? | ● | ● |
| Porites columnaris[2] | | ● | ● |
| Porites compressa[2,3,5] | ? | ● | ● |
| Porites lutea[2,5] | ? | ? | ● |
| Porites nodifera[5] | ? | ? | ● |
| Porites solida[2,5] | ? | ● | ● |
| Seriatopora caliendrum[2,5] | ? | ● | ● |
| Seriatopora hystrix[2] | | ● | ● |
| Siderastrea savignyana[1,2,5] | ● | | ● |
| Stylocoeniella guentheri[1] | ● | | ● |
| Stylophora pistillata[2,3,5] | ? | ? | ● |
| Stylophora danae[5] | ? | ? | ● |

(Continued)

| Pleistocene corals | Middle Pleistocdene | Late Pleistocene | Modern |
|---|---|---|---|
| *Stylophora kuehlmanni*[5] | ? | ? | ■ |
| *Stylophora subseriata*[5] | ? | ? | ■ |
| *Turbinaria mesenterina*[2] | | ■ | ■ |
| *Turbinaria peltata*[2,3,5] | ? | ? | |

**Notes:**

[1] Reported in this study.

[2] Reported in *Dullo (1990)*.

[3] Reported from *Bruggemann et al. (2004)*.

[4] Reported in *Al-Rifaiy & Cherif (1988)*.

[5] Reported from *El-Sorogy (2008)*. *El-Sorogy (2008)* does not distinguish between Middle and Late Pleistocene, so taxa reported therein are represented with a gray line across both Middle and Late Pleistocene and marked with ? unless they have been confirmed as Middle or Late Pleistocene in a second publication. Black bar indicates species is present for the time period.

[*] Taxa were originally reported as a synonym of updated species name listed here. In order of appearance they are: *Acropora abrotanoides* originally reported as *Acropora tutuilensis* (new placement following *Hoeksema, 2013*); *Astrea curta* originally reported as *Montastrea curta* (new placement following *Hoeksema, 2014*); *Cantharellus doederleini* originally reported as *Cycloseris doederleini* (new placement following *Hoeksema, 2016*); *Coelastrea aspera* originally reported as *Goniastrea aspera* (new placement following *Hoeksema, 2014*); *Cycloseris costulata* originally reported s as *Cycloseris marginata* (new placement following *Hoeksema, 2014*); *Cycloseris tenuis* originally reported as *Cycloseris erosa* (new placement following *Hoeksema, 2014*); all *Dipsastraea* species originally reported as genus *Favia* (new placement following *Hoeksema, 2017*); *Favites rotundata* originally reported as *Favia rotundata* (new placement following *Hoeksema, 2014*); *Goniastrea stelligera* originally reported as *Favia stelligera* (new placement following *Hoeksema, 2014*); *Goniopora pedunculata* originally reported as *Goniopora minor* (new placement following *Hoeksema, 2014*); *Leptoseris yabei* originally reported as *Pavona yabei* (new placement following *Hoeksema, 2014*); *Lobophyllia erythraea* originally reported as *Symphyllia erythraea* (new placement following *Hoeksema, 2016*); *Oulophyllia bennettae* originally reported as *Favites bennettae* (new placement following *Hoeksema, 2014*); *Paragoniastrea australensis* originally reported as *Goniastrea australensis* (new placement following *Hoeksema, 2017*); *Paramontastrea peresi* originally reported as both *Favites peresi* and *Goniastrea peresi* (new placement following *Hoeksema, 2014*); *Porites rus* originally reported as both *Porites iwayamaensis* and *Porites undulata* (new placement following *Hoeksema, 2014*).

composed of 90% endogenous material (*Dudley, 1996*). Non-marine sand was a very poorly sorted mix of sand, gravel, and pebble-sized grains of quartz-rich rock, and lacked fragments of marine invertebrate tests (see Fig. 3C). Carbonate mud was well-sorted, composed primarily of silt-sized grains, and reacted with dilute HCl (see Fig. 3D).

## Taxonomy

A total of 52 Scleractinian species in 26 genera and nine families (Figs. 4–20) were identified from all transects. *Budd et al. (2012)* were unable to place the genus *Leptastrea*, so it is classed as *incertae familiae* in this study. Thirty-six of these were identified with a high degree of confidence and nine were given tentative identifications due to poor preservation. Seven were identified to the genus level and two were only identifiable to the family level. Specimens of the genera *Acropora* and *Porites* were not identified to the species level because they could not be consistently identified throughout transects, and were instead treated as a single taxonomic group, leading to an underestimation of true diversity. Just three families, Merulinidae, Poritidae, and Euphylliidae, accounted for 87.8% of all identified corals, and 48% belonged to the family Poritidae.

The taxonomic figures provided (Figs. 4–20) include examples of corallite detail and growth form whenever possible, examples from the Middle and Late Pleistocene when species occur on both terraces, and thin sections of the most poorly preserved taxa.

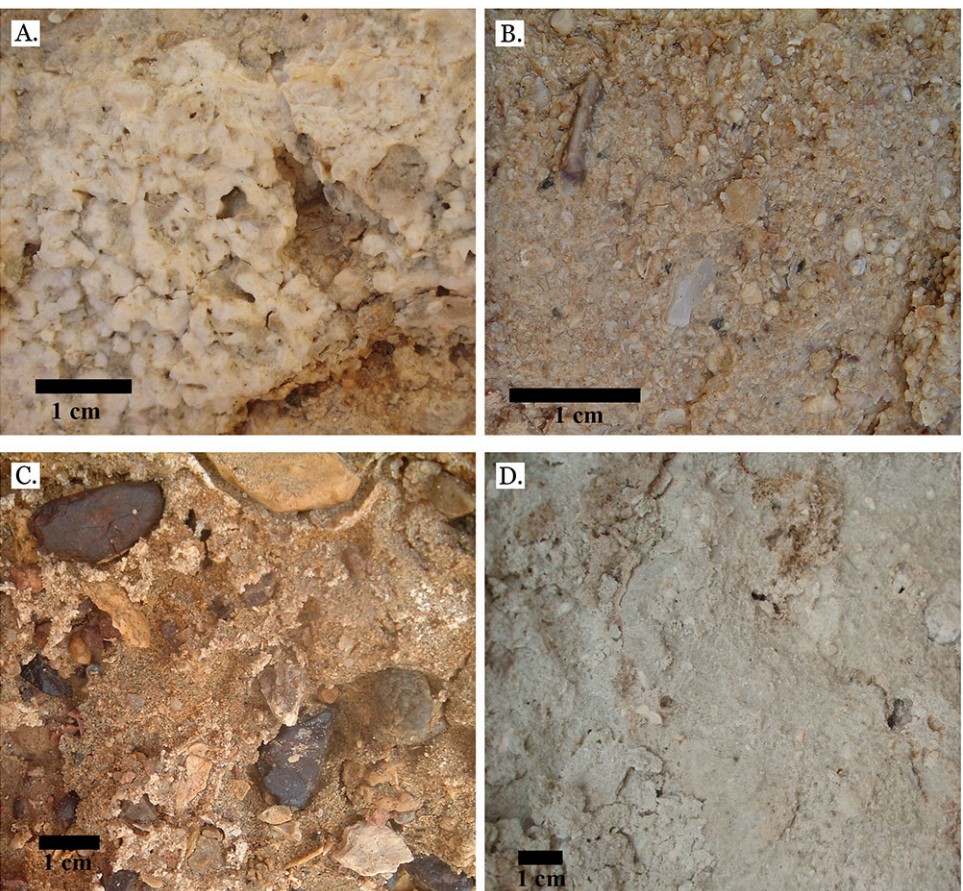

**Figure 3 Sediment types.** (A) Coralline algae, rhodolithic form, from middle Pleistocene Wadi Wizr; (B) Marine sand from Late Pleistocene Wadi Wizr; (C) Non-marine sand from Middle Pleistocene Wadi Gawasis; (D) Carbonate mud from Middle Pleistocene Wadi Wizr.

This is to show the variability in coral appearance due to preservation and to provide a useful reference for identifying fossil corals in the field.

Forty-three of the 46 reliably identified coral species found on Middle and Late Pleistocene reefs are found in the modern Red Sea (*DiBattista et al., 2015*; *Veron, 2000*), and the specimens that I could only identify to family or genus level show no characters suggesting that they are either new or locally extinct species; in every case, they resemble one or more species found on the modern Red Sea but simply lack enough detail for confident identification.

*Favites micropentagonus* and *Pavona minuta* are the two species found on Pleistocene transects in this study that are not reported from the modern Red Sea. *Favites micropentagonus* is reported from the Middle Pleistocene, but not the Late Pleistocene or modern Red Sea and *Pavona minuta* is reported from the Late Pleistocene, but not the Middle Pleistocene or modern Red Sea. *Astrea curta*, *Pavona decussata*, and *Stylocoeniella guentheri* are the only three taxa to occur on all Middle Pleistocene terraces and in the modern Red Sea, but not on any Late Pleistocene terraces. Five other species are only found on Middle Pleistocene reefs in this study, but they have been reported from

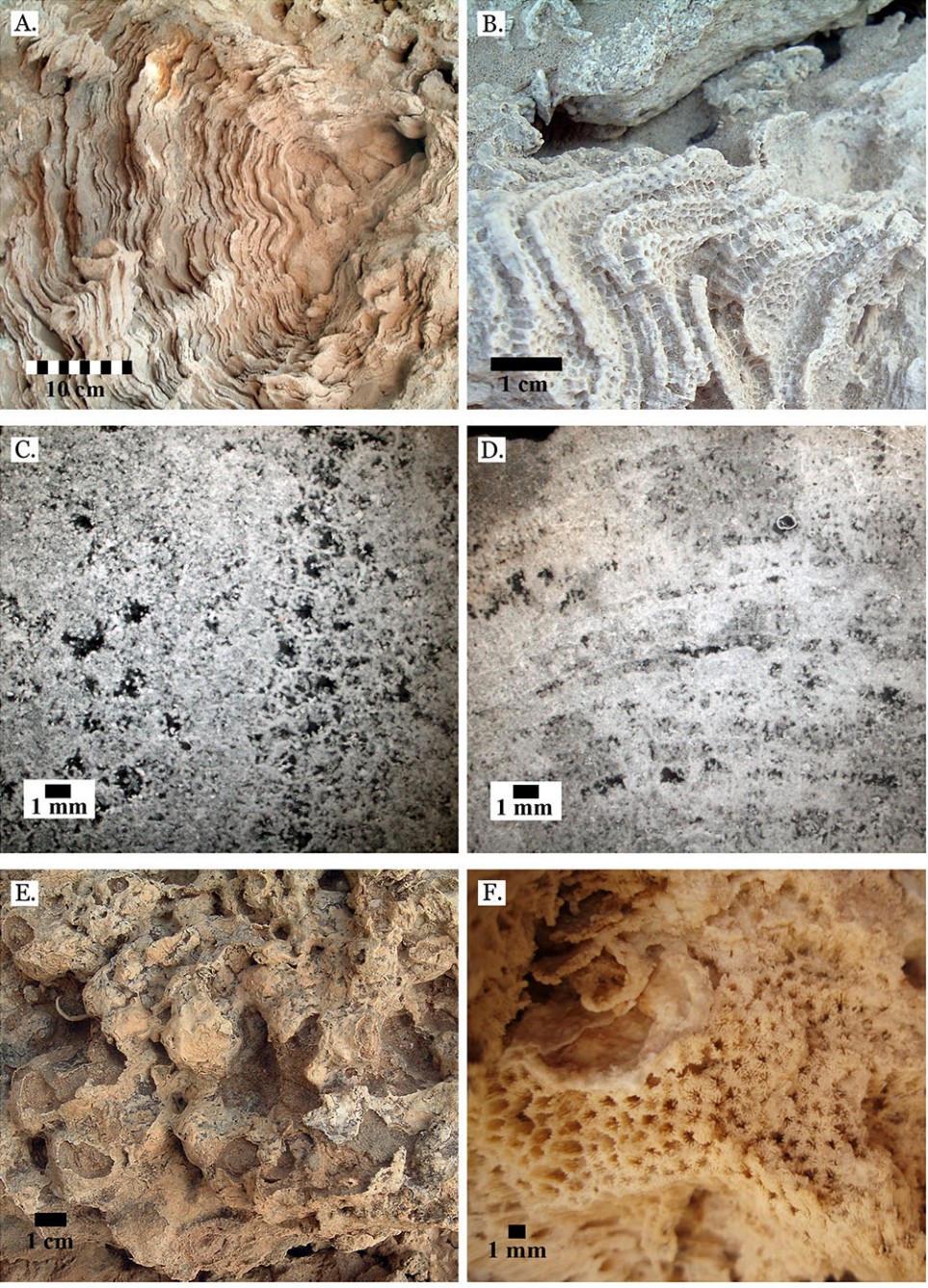

**Figure 4 Family Poritidae.** (A) Massive *Porites* sp. from Middle Pleistocene Wadi Gawasis; (B) Close up of massive *Porites* sp. from Middle Pleistocene Sharm Al Arab; (C) Latitudinal thin section of *Porites* sp. from Middle Pleistocene Sharm Al Arab, UCMP# 557250; (D) Longitudinal thin section of *Porites* sp. from Middle Pleistocene Sharm Al Arab, UCMP# 557250; (E) Branching *Porites* sp. from Middle Pleistocene Wadi Gawasis; (F) *Porites* sp. from Middle Pleistocene Wadi Wizr.

Late Pleistocene reefs in the Red Sea literature. Nearly 48% of species (22) are only reported from the Late Pleistocene and Modern Red Sea, but not from the Middle Pleistocene. Table 1 gives the age occurrences of all fossil coral taxa reported from the Red Sea basin here and in previous studies.
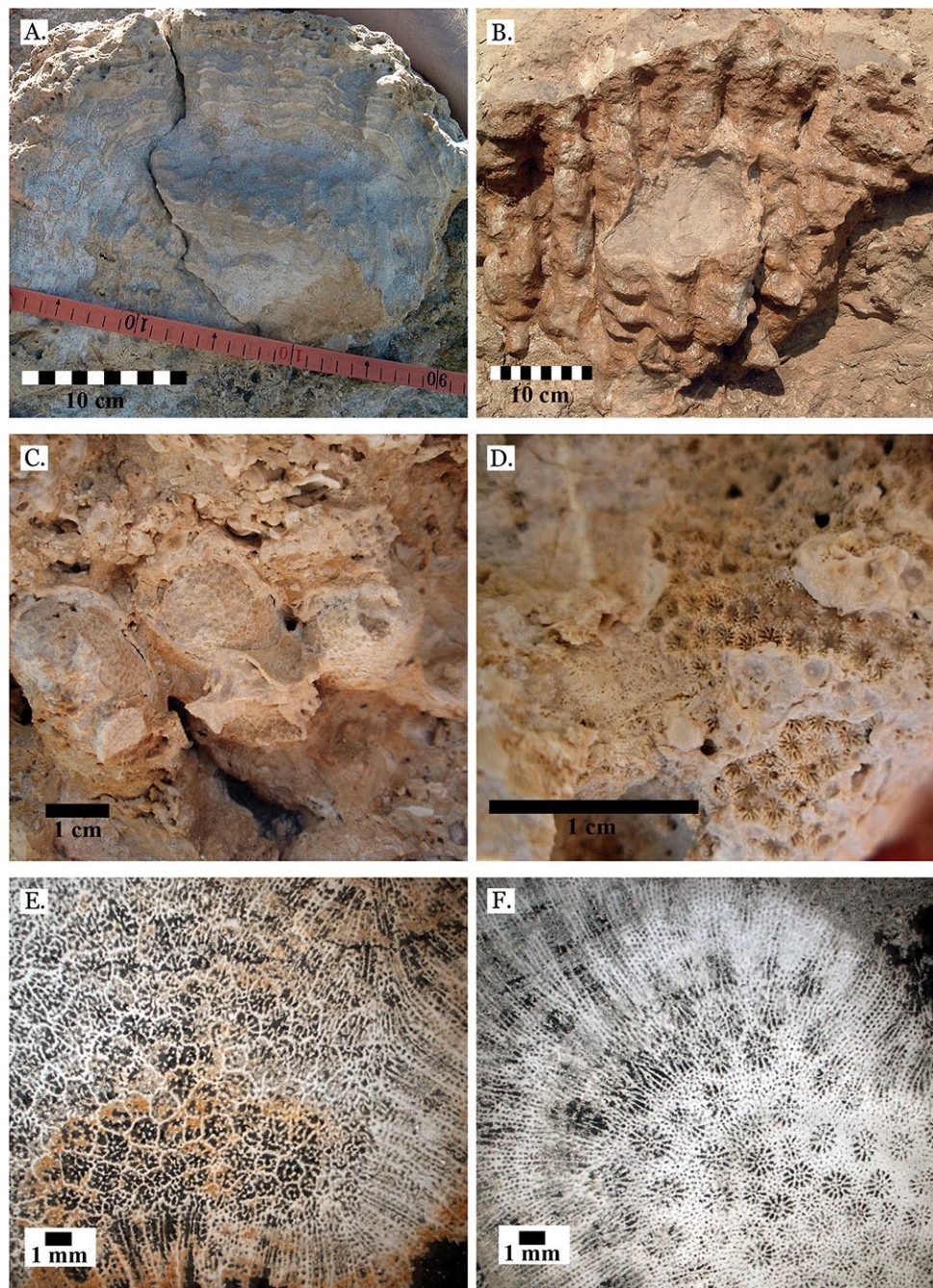

**Figure 5 Family Poritidae.** (A) Massive *Porites* sp. from Late Pleistocene Wadi Gawasis; (B) Robust branching *Porites* sp. from Late Pleistocene Wadi Wizr; (C) Branches of *Porites* sp. from Late Pleistocene Wadi Wizr; (D) *Porites* sp. from Late Pleistocene Sharm Al Arab; (E) Thin section of *Porites* sp. from the Late Pleistocene; (F) Thin section of *Porites* sp. from the Late Pleistocene.

## Middle Pleistocene terraces

### Wadi Wizr

I made 130 identifications of taxa or sediment type on the Wadi Wizr Middle Pleistocene terrace transect, including nine coral species. Coral cover was 60%, coralline algae 12.3%,

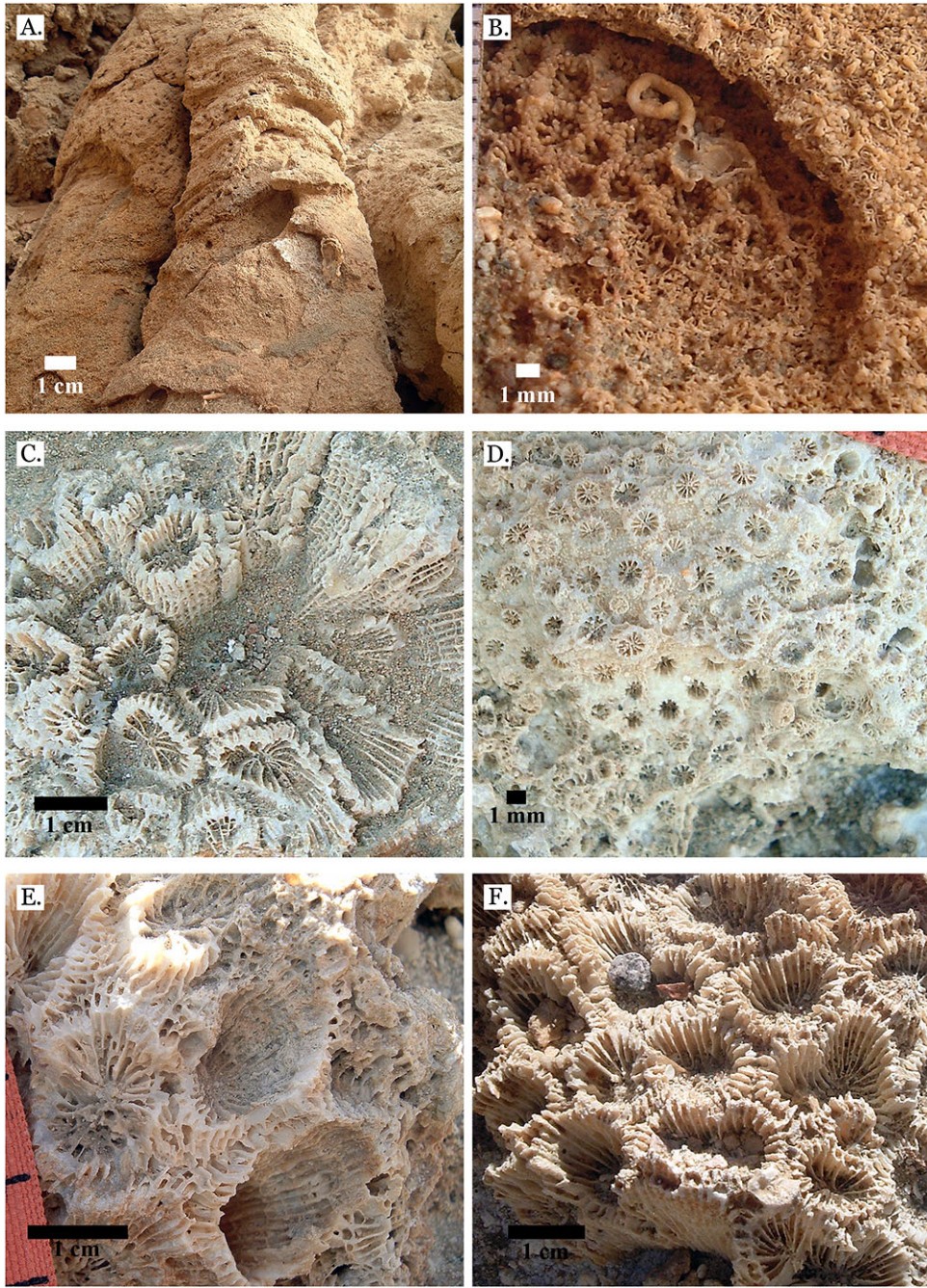

**Figure 6 Family Poritidae.** (A) *Goniopora* sp. from Middle Pleistocene Wadi Gawasis, (B) *Goniopora* sp. from Middle Pleistocene Wadi Gawasis. Family Merulinidae. (C) *Caulastrea tumida* from Late Pleistocene Sharm Al Arab; (D) *Cyphastrea microphthalma* from Late Pleistocene Wadi Gawasis; (E) *Dipsastraea favus* from Late Pleistocene Sharm Al Arab; (F) *Dipsastraea matthai* from Late Pleistocene Wadi Gawasis.

marine sand 9%, and carbonate mud was 8.5% (see Table 2 for a summary of all terraces). The most abundant coral were massive *Porites* spp. (65.4% of coral identifications). *Goniastrea stelligera* (20.5%) was the second most abundant, followed by *Stylocoeniella guentheri* (6.4%). *Cyphastrea microphthalma*, *Gardineroseris planulata*, *Astrea curta*, *Platygyra sinensis*,

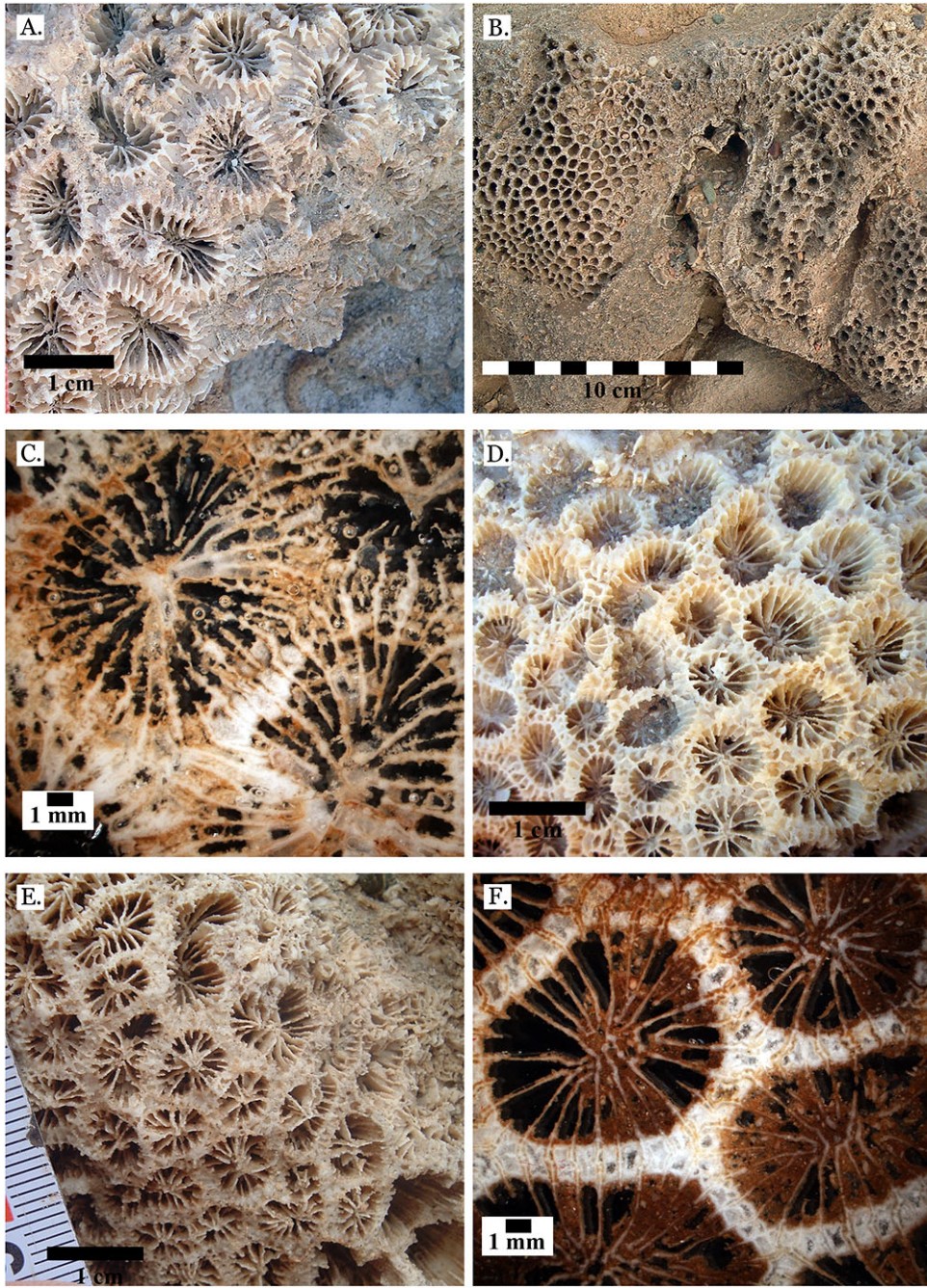

**Figure 7 Family Merulinidae.** (A) *Dipsastraea pallida* from Late Pleistocene Sharm Al Arab; (B) *Dipsastraea pallida* from Middle Pleistocene Wadi Gawasis; (C) Latitudinal thin section of *Dipsastraea pallida* from Middle Pleistocene Wadi Gawasis, UCMP# 557248; (D) *Dipsastraea speciosa* from Late Pleistocene Wadi Gawasis; (E) *Dipsastraea speciosa* from Middle Pleistocene Wadi Gawasis; (F) Latitudinal thin section of *Dipsastraea speciosa* Middle Pleistocene Wadi Gawasis, UCMP# 557247.

*Dipsastraea* sp., and *Goniastrea* sp., all accounted for 1.8% (see Table 3 for species abundances on all terraces). The Middle Pleistocene Wadi Wizr terrace had the second lowest Shannon–Wiener index (H) (1.57) and the fourth lowest Simpson's diversity index (0.732).

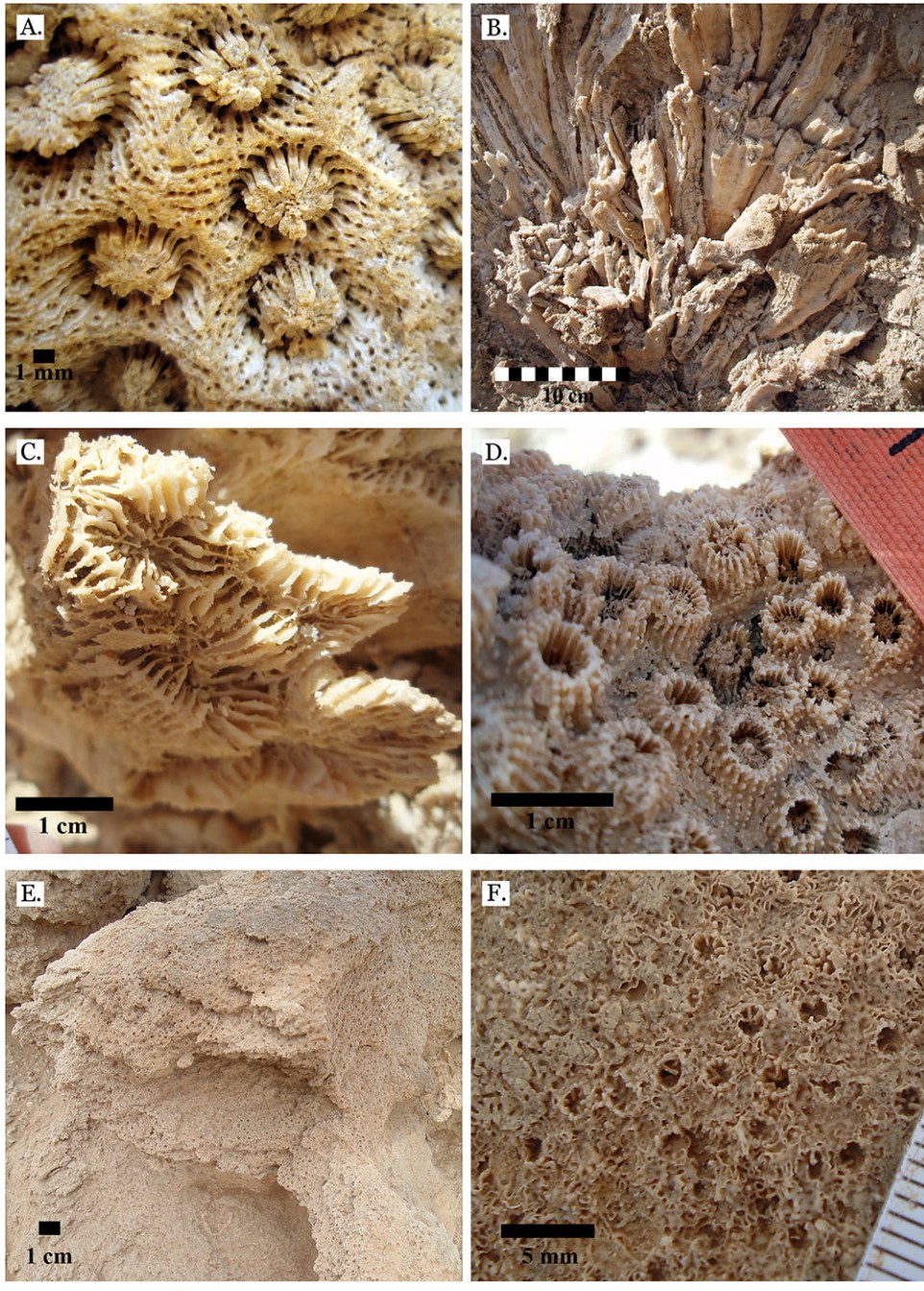

**Figure 8 Family Merulinidae.** (A) *Dipsastraea* sp. mold from Middle Pleistocene Wadi Wizr; (B) *Erythrastrea flabellata* from Late Pleistocene Wadi Wizr; (C) *Erythrastrea flabellata* from Late Pleistocene Wadi Wizr; (D) *Echinopora gemmacea* from Late Pleistocene Sharm Al Arab; (E) *Echinopora gemmacea* from Middle Pleistocene Wadi Gawasis; (F) *Echinopora gemmacea* from Middle Pleistocene Wadi Gawasis.

## Wadi Gawasis

One hundred seventy-nine identifications were made of taxa or sediment type on the Wadi Gawasis Middle Pleistocene terrace. These included 24 coral species. Coral cover was 78%, non-marine sand was 12.9% of the transect points, and marine sand 9% (Table 2).

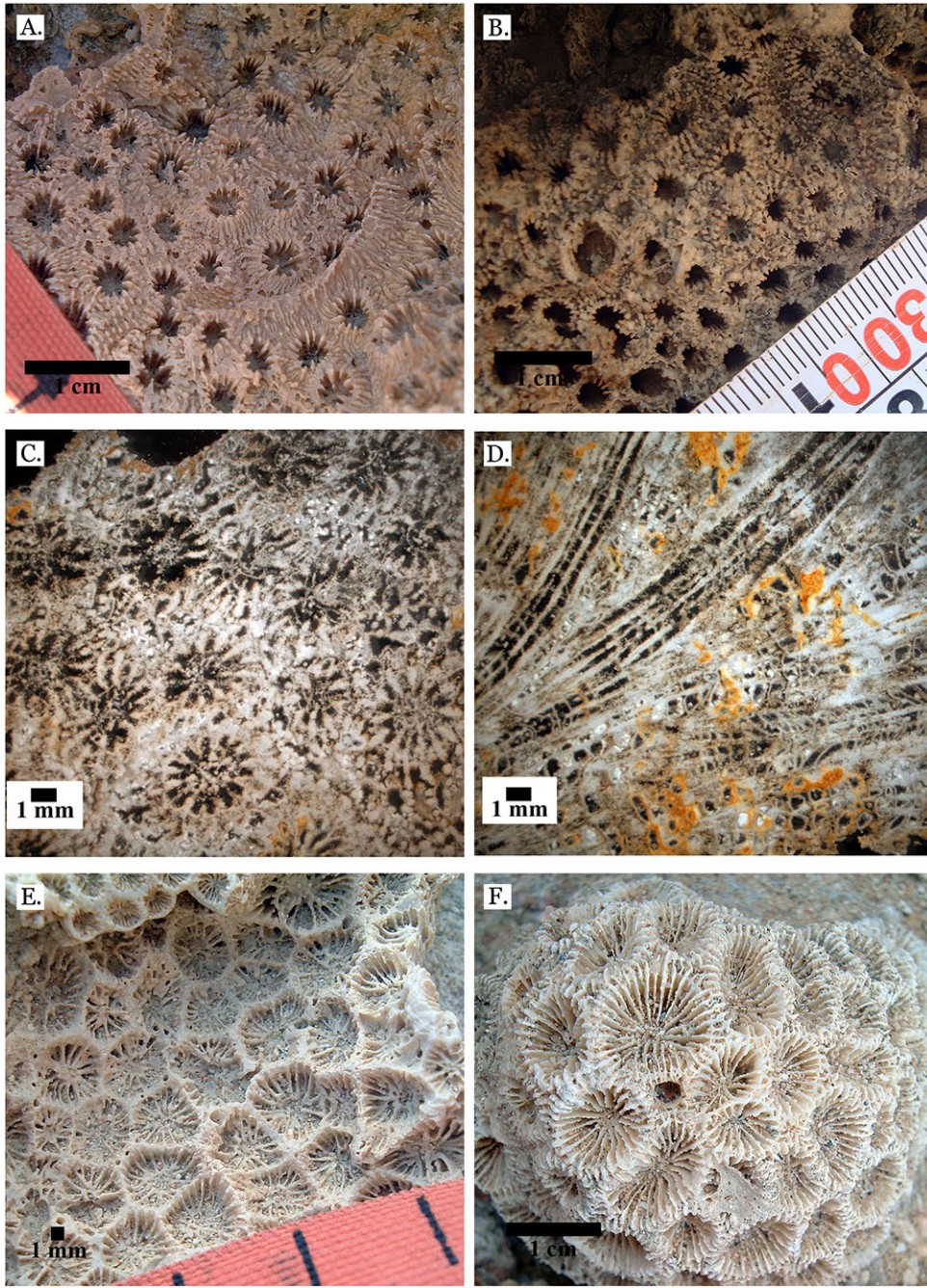

**Figure 9 Family Merulinidae.** (A) *Echinopora lamellosa* from Late Pleistocene Wadi Gawasis; (B) *Echinopora lamellosa* from Middle Pleistocene Wadi Gawasis; (C) Latitudinal thin section of *Echinopora lamellosa* from the Middle Pleistocene; (D) Longitudinal thin section of *Echinopora lamellosa* from the Middle Pleistocene; (E) *Favites abdita* from Late Pleistocene Sharm Al Arab; (F) *Favites flexuosa* from Late Pleistocene Wadi Gawasis.

The most abundant coral were primarily branching *Porites* spp. (42.9% of coral identifications). *Galaxea fascicularis* was the second most abundant at 9.3%, and *Echinopora lamellosa* was the third most abundant at 8.6%. *Favites micropentagonus* and

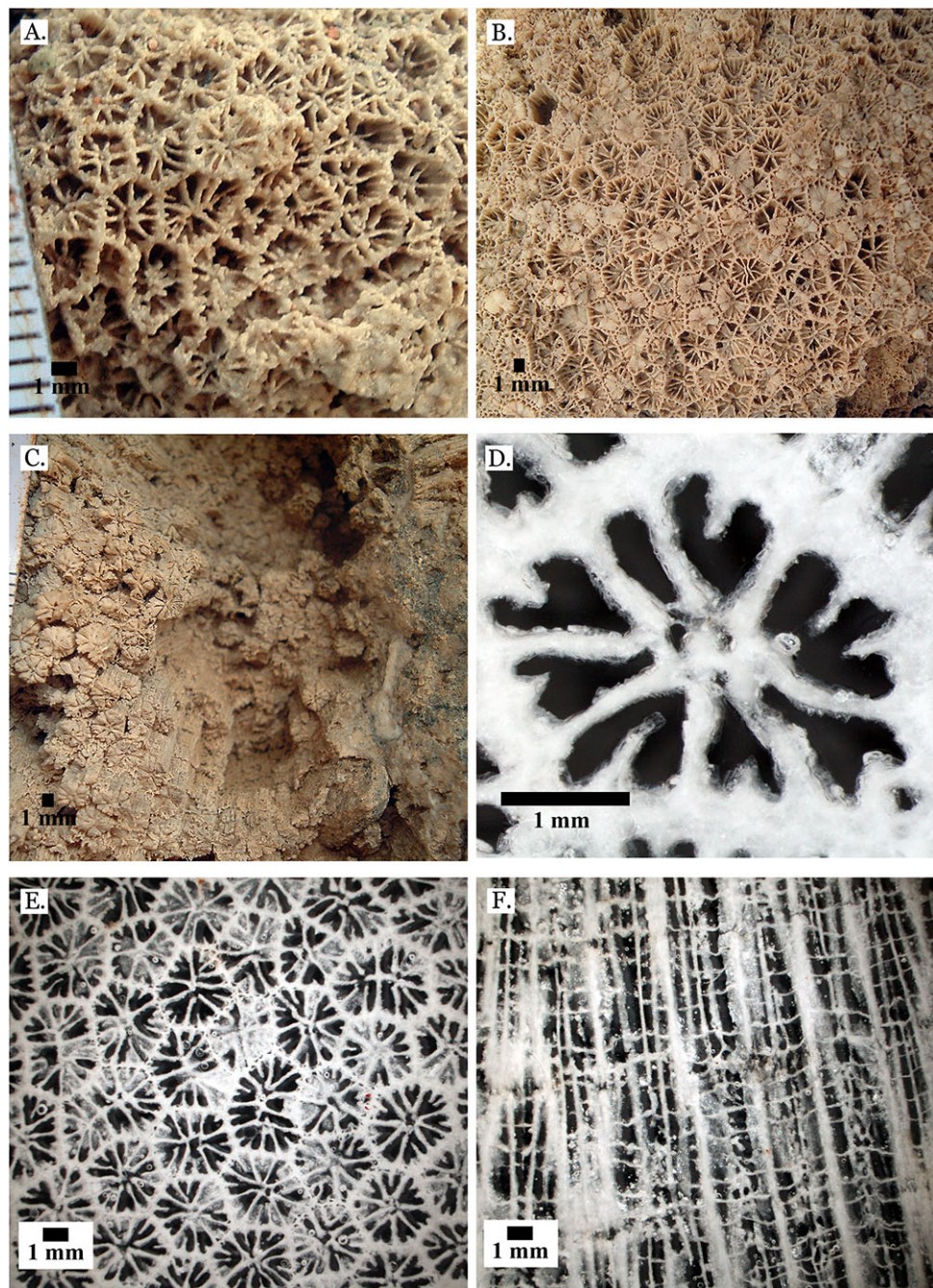

**Figure 10 Family Merulinidae.** (A) *Favites micropentagonus* from Middle Pleistocene Wadi Gawasis; (B) *Favites micropentagonus* from Middle Pleistocene Wadi Gawasis; (C) *Favites micropentagonus* mold from Middle Pleistocene Wadi Gawasis; (D) Latitudinal thin section of *Favites micropentagonus* from Middle Pleistocene Wadi Gawasis, UCMP# 557239; (E) Latitudinal thin section of *Favites micropentagonus* from Middle Pleistocene Wadi Gawasis, UCMP# 557239; (F) Longitudinal thin section of *Favites micropentagonus* from Middle Pleistocene Wadi Gawasis, UCMP# 557239.

*Goniopora* sp. were 7.1% each, and *Pocillopora damicornis*, *Stylocoeniella guentheri*, and an unidentified Faviidae were 2.9%. *Goniastrea retiformis* and *Astrea curta* accounted for 2.1% each. *Goniastrea stelligera*, *Leptastrea bottae*, and *Pavona* c.f. *bipartita* were 1.4%

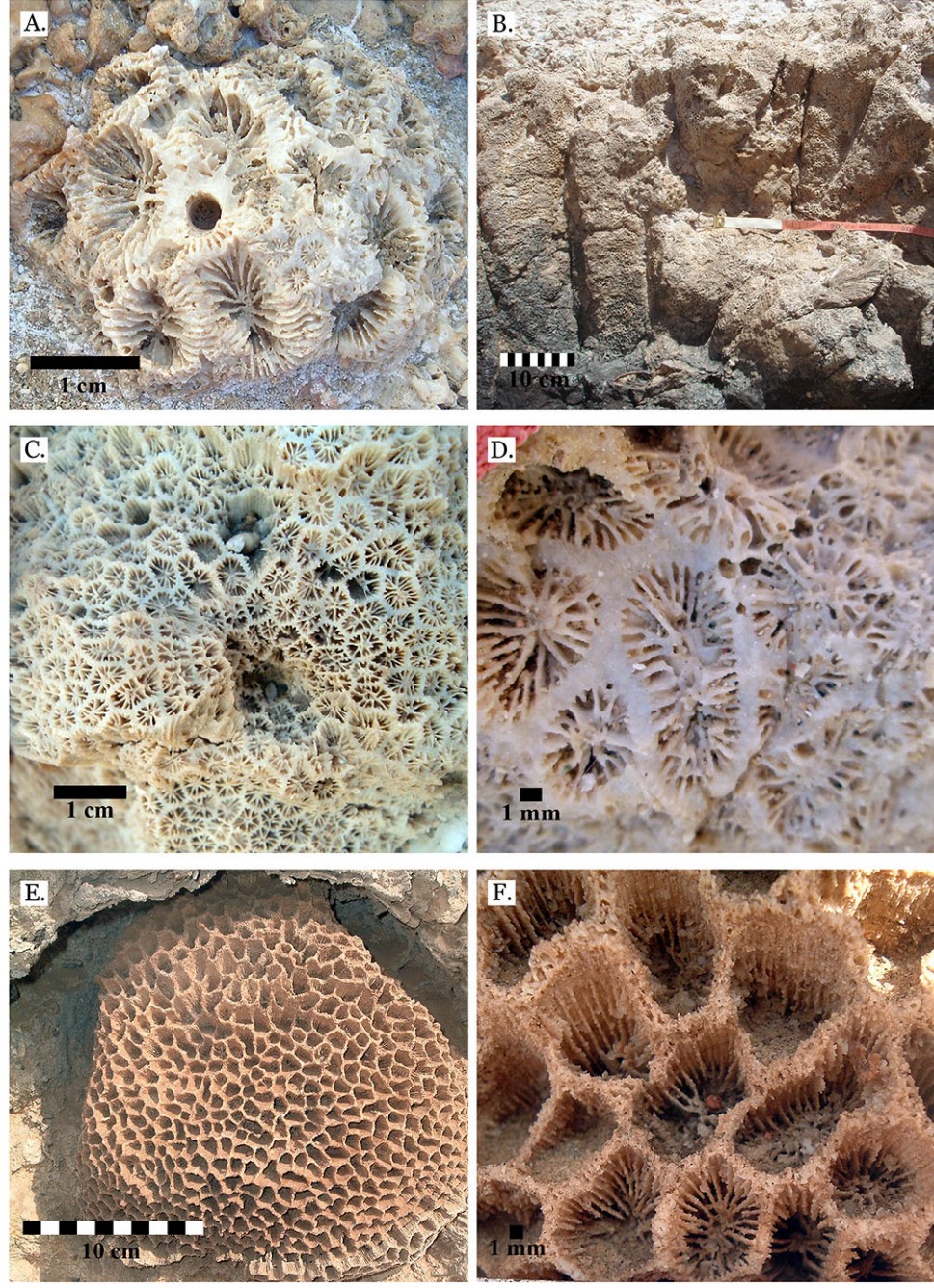

**Figure 11 Family Merulinidae.** (A) *Favites paraflexuosa* from Late Pleistocene Wadi Gawasis; (B) *Favites pentagona* from Late Pleistocene Sharm Al Arab; (C) *Favites pentagona* from Late Pleistocene Wadi Wizr; (D) *Favites vasta* from Late Pleistocene Wadi Gawasis; (E) *Favites* sp. from Middle Pleistocene Wadi Gawasis; (F) *Favites* sp. from Middle Pleistocene Wadi Gawasis.

each, while *Acropora* sp., *Cyphastrea seralia*, *Dipsastraea pallida*, *Dipsastraea speciosa*, *Echinopora gemmacea*, *Gardineroseris planulata*, *Gyrosmilia interrupta*, *Pavona decussata*, *Siderastrea savignyana*, *Favites* sp., and *Goniastrea* sp., all accounted for fewer than 1%

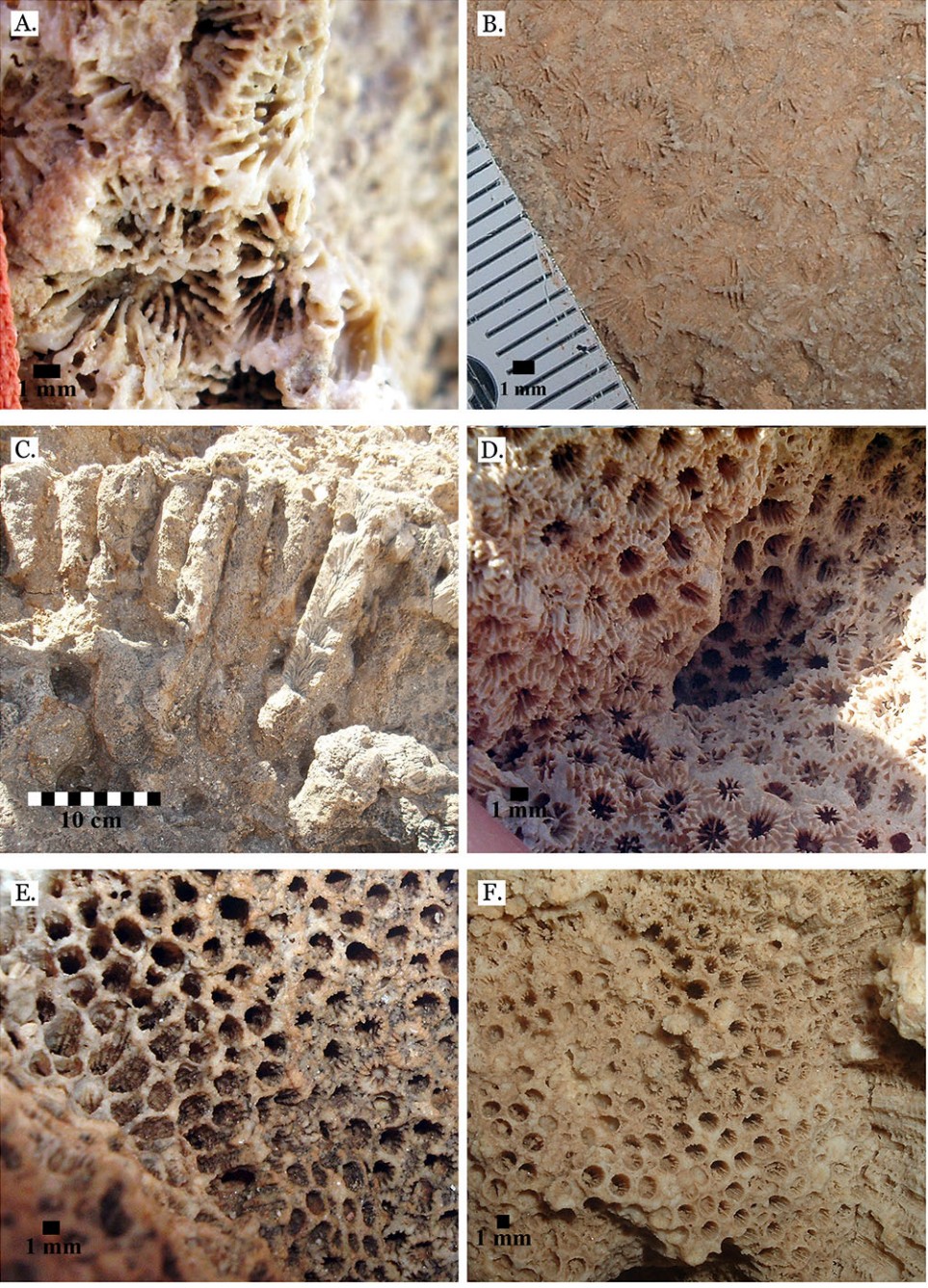

**Figure 12 Family Merulinidae.** (A) *Goniastrea pectinata* from Late Pleistocene Wadi Gawasis; (B) *Goniastrea retiformis* from Middle Pleistocene Wadi Gawasis; (C) *Goniastrea stelligera* from Late Pleistocene Wadi Gawasis; (D) *Goniastrea stelligera* from Late Pleistocene Sharm Al Arab; (E) *Goniastrea stelligera* from Middle Pleistocene Wadi Wizr; (F) Very poorly preserved *Goniastrea stelligera* from Middle Pleistocene Wadi Wizr.

of coral identifications (Table 3). Wadi Gawasis had the highest diversity of the three Middle Pleistocene sites and the third highest diversity of all the terraces with an H of 2.42 and a Simpson's diversity index of 0.844.

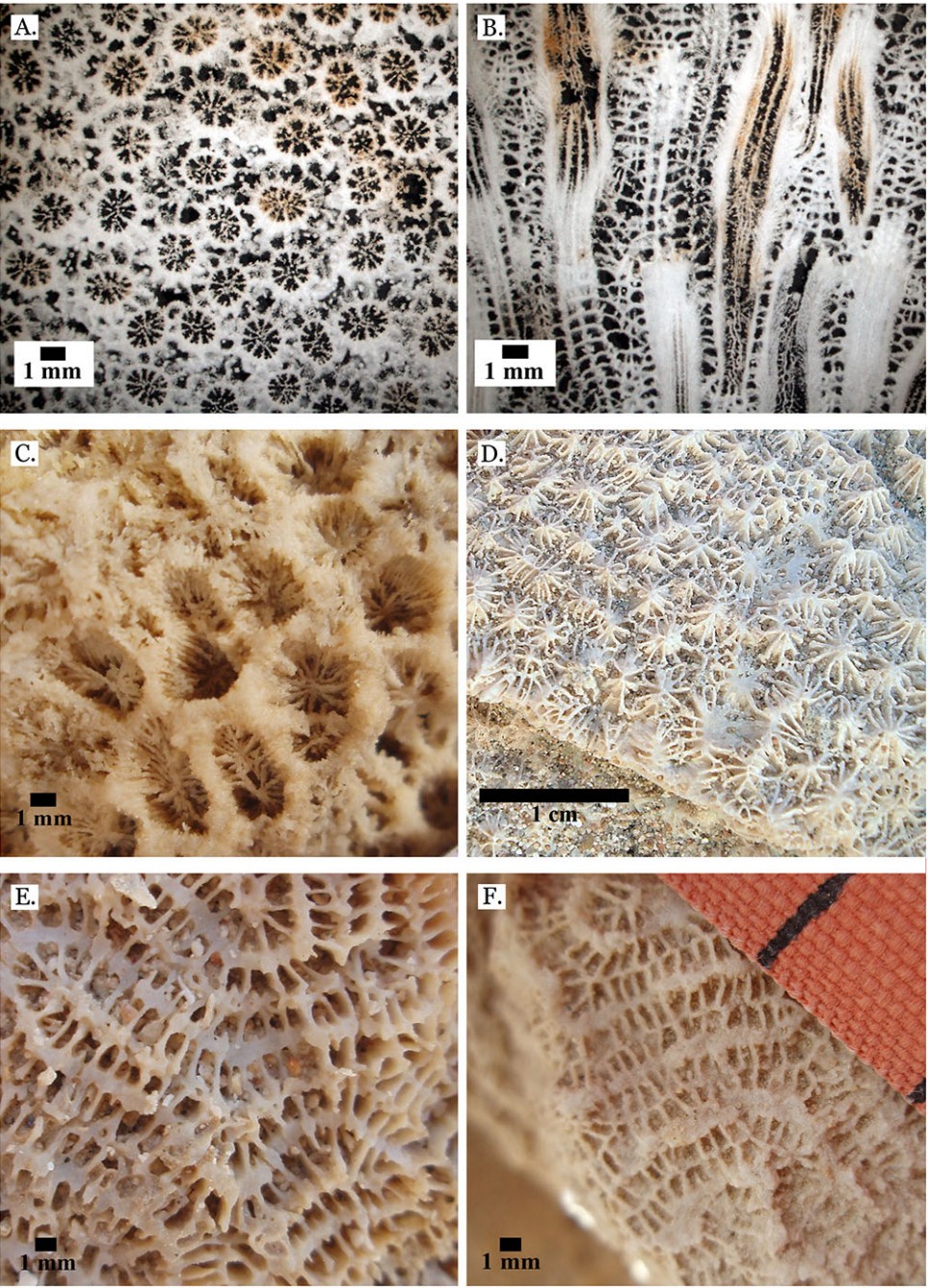

**Figure 13 Family Merulinidae.** (A) Latitudinal thin section of *Goniastrea stelligera* from Middle Pleistocene Wadi Wizr, UCMP# 557194; (B) Longitudinal thin section of *Goniastrea stelligera* from Middle Pleistocene Wadi Wizr, UCMP# 557194; (C) *Goniastrea* sp. from Middle Pleistocene Wadi Wizr; (D) *Hydnophora microconus* from Late Pleistocene Wadi Gawasis; (E) *Leptoria Phrygia* from Late Pleistocene Wadi Gawasos; (F) *Leptoria phrygia* from Middle Pleistocene Sharm Al Arab.

### Sharm el Arab

I made 110 identifications of taxa or sediment type on the Sharm el Arab Middle Pleistocene terrace transect, including seven coral species. Coral cover was 68.2%, marine sand was 28.2%, and coralline algae was 3.6% (Table 2). The most abundant coral

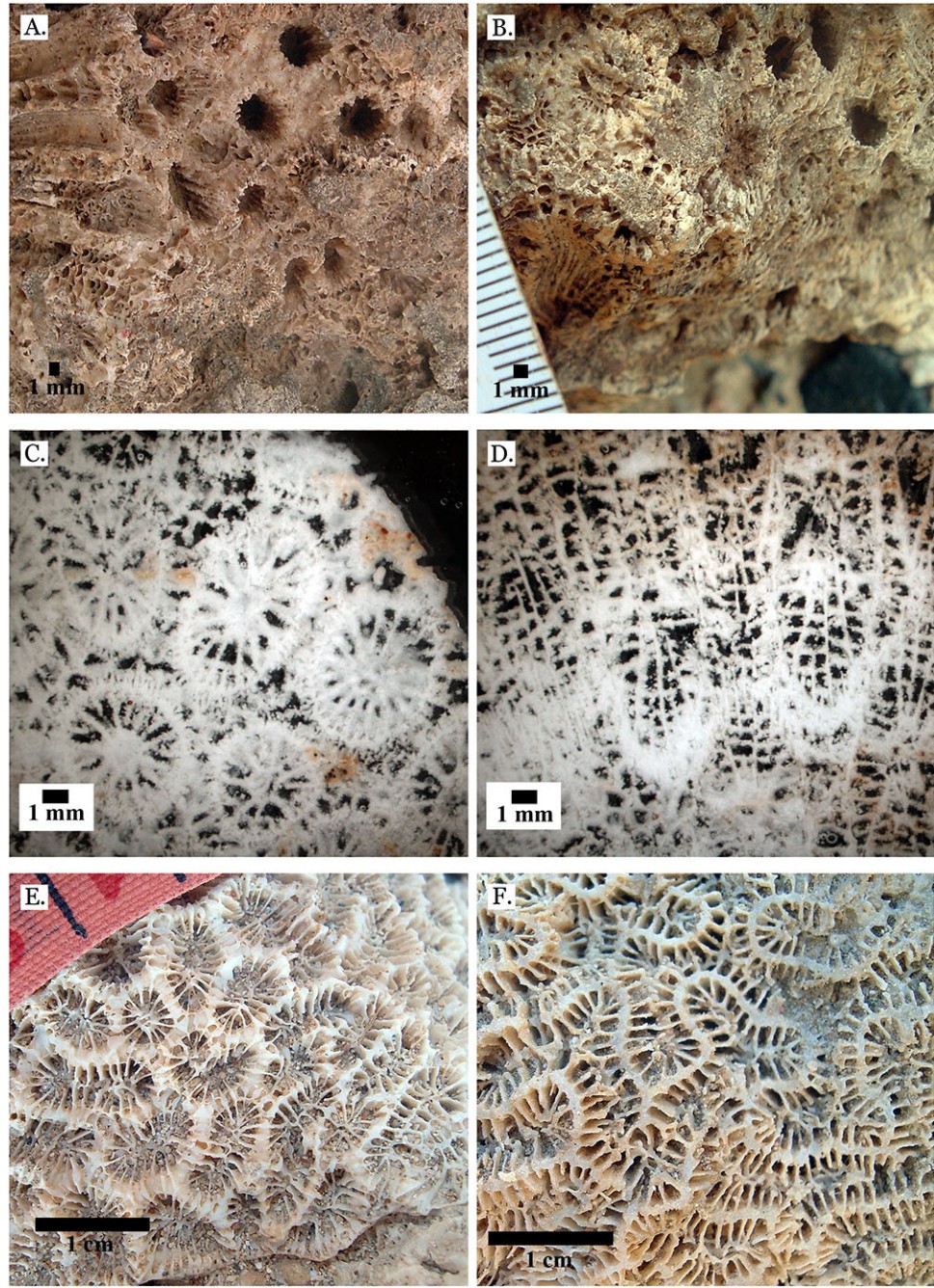

**Figure 14 Family Merulinidae.** (A) *Astrea curta* from Middle Pleistocene Wadi Gawasis; (B) Very poorly preserved *Astrea curta* from Middle Pleistocene Wadi Gawasis; (C) Latitudinal thin section of *Astrea curta* from Middle Pleistocene; (D) Longitudinal thin section of *Astrea curta* from Middle Pleistocene; (E) *Platygyra acuta* from Late Pleistocene Wadi Gawasis; (F) *Platygyra daedalea* from Late Pleistocene Sharm Al Arab.

were massive *Porites* spp. (86.7% of coral identifications). *Echinopora lamellosa*, *Astrea curta*, *Stylocoeniella guentheri*, and an unidentifiable Agariciidae were 2.7% each, and *Leptoria phrygia* and *Pocillopora damicornis* accounted for 1.3% each (Table 3). The Middle Pleistocene terrace at Sharm el Arab had the lowest diversity of all eight terraces with an H of 1.16 and a Simpson's diversity index of 0.569.

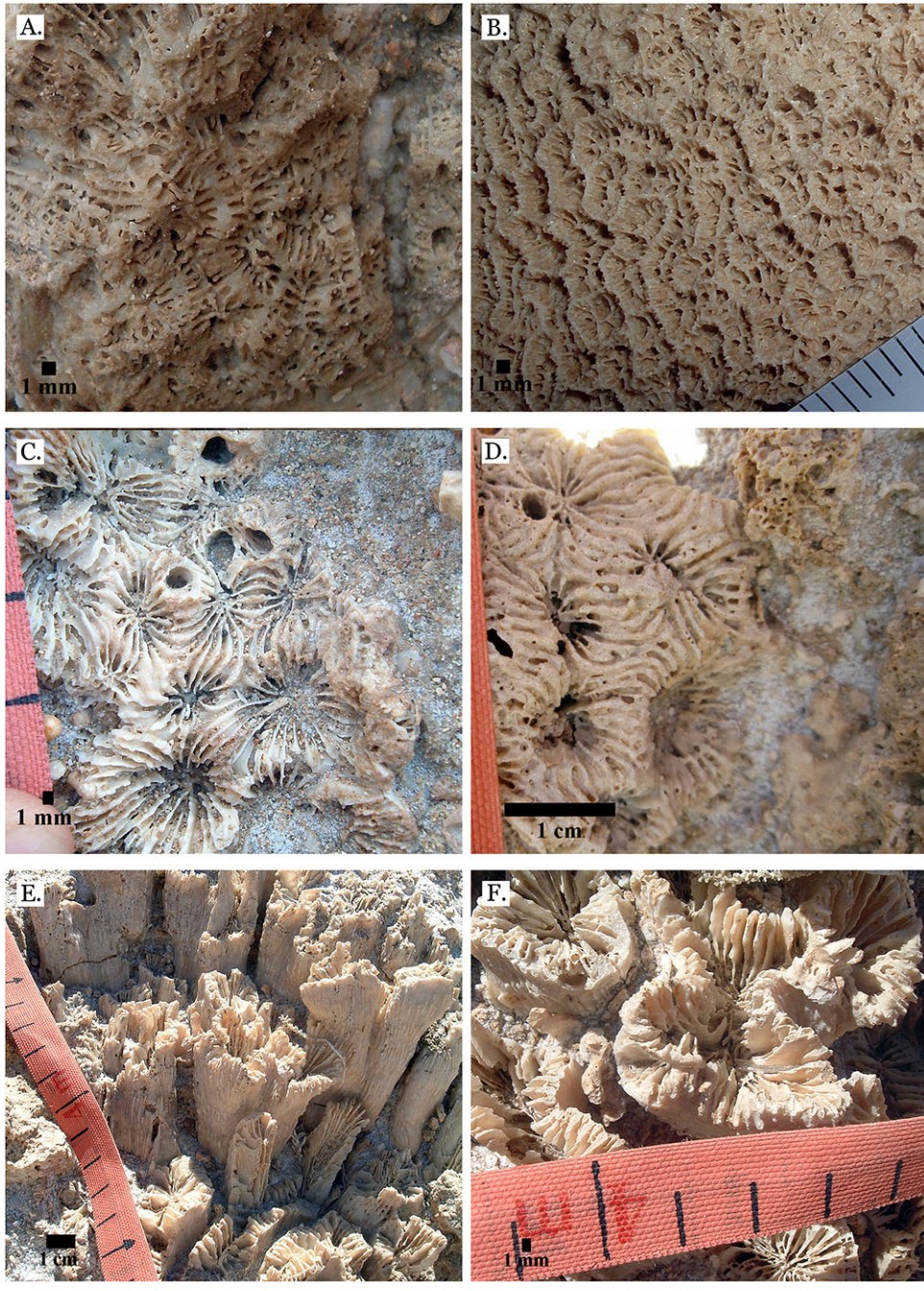

**Figure 15 Family Merulinidae.** (A) *Platygyra lamellina* from Late Pleistocene Wadi Gawasis; (B) *Platygyra sinensis* from Middle Pleistocene Wadi Wizr. Family Lobophylliidae. (C) *Acanthastrea echinata* from Late Pleistocene Wadi Gawasis; (D) *Acanthastrea rotundoflora* from Late Pleistocene Wadi Gawasis; (E) *Lobophyllia hemprichii* from Late Pleistocene Wadi Gawasis; (F) *Lobophyllia hemprichii* from Late Pleistocene Wadi Gawasis.

## Late Pleistocene terraces

### *Wadi Wizr—upper terrace*

One hundred twenty-nine identifications were made of taxa or sediment on the Wadi Wizr Late Pleistocene upper terrace transect, including 20 coral species. Coral cover

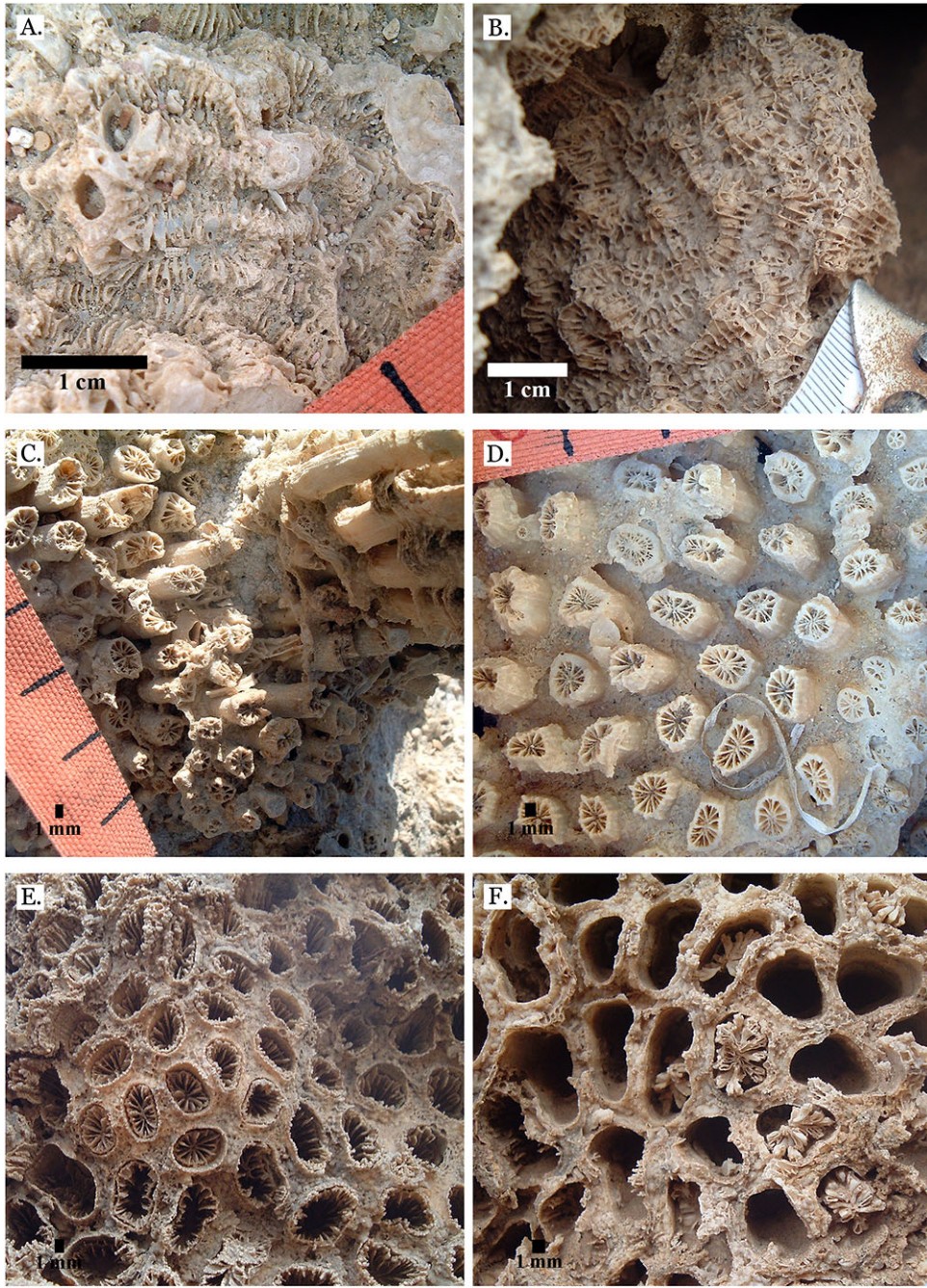

**Figure 16 Family Euphylliidae.** (A) *Gyrosmilia interrupta* from Late Pleistocene Sharm Al Arab; (B) *Gyrosmilia interrupta* from Middle Pleistocene Wadi Gawasis; (C) *Galaxea fascicularis* from Late Pleistocene Sharm Al Arab; (D) Young *Galaxea fascicularis* on edge of colony from Late Pleistocene Wadi Gawasis, easily mistaken for *Galaxea astreata*; (E) *Galaxea fascicularis* from Middle Pleistocene Wadi Gawasis; (F) Very poorly preserved *Galaxea fascicularis* from Middle Pleistocene Wadi Gawasis.

was 97.7%, marine sand was 2%, and coralline algae 1% (Table 2). The most abundant coral were primarily branched *Porites* spp. (50% of coral identifications). *Galaxea fascicularis* (19%) was the second most abundant, followed by *Pocillopora damicornis*

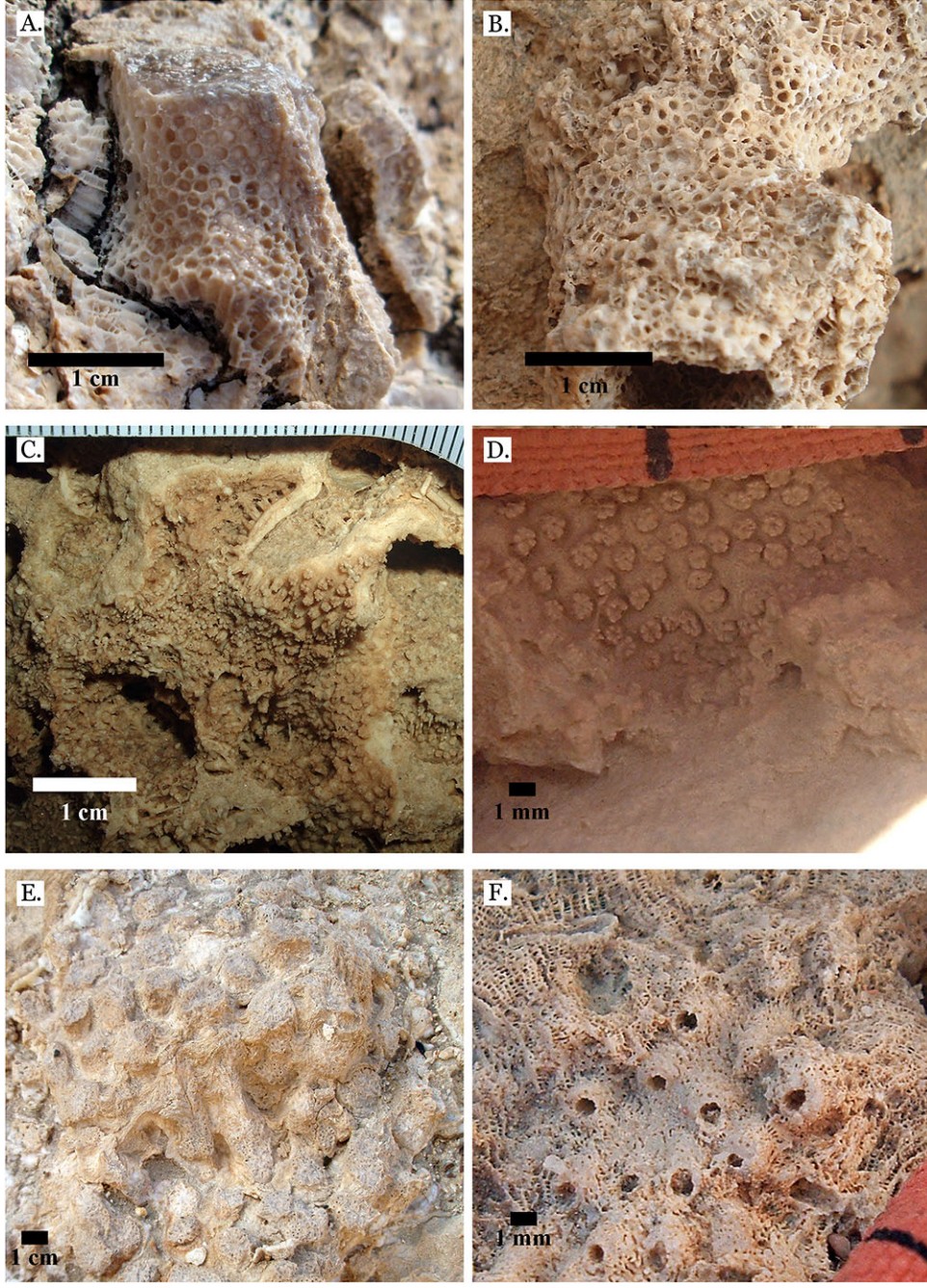

**Figure 17 Family Pocilloporidae.** (A) *Pocillopora damicornis* from Late Pleistocene Wadi Wizr; (B) *Pocillopora damicornis* from Middle Pleistocene Wadi Gawasis. Family Acroporidae. (C) *Acropora* sp. from Late Pleistocene Wadi Wizr; (D) *Acropora* sp. from Late Pleistocene Sharm Al Arab. Family Astrocoeniidae; (E) *Stylocoeniella guentheri* mold from Middle Pleistocene Wadi Wizr; (F) *Stylocoeniella guentheri* mold from Middle Pleistocene Sharm Al Arab.

(5.6%). *Goniastrea stelligera*, *Dipsastraea speciosa*, and *Favites pentagona* were 3.2% each, *Acropora* sp. and *Dipsastraea pallida* were 2.4% each, and *Favites vasta* and *Hydnophora microconos* were 1.6% each. *Dipsastraea favus*, *Echinopora gemmacea*, *Echinopora*

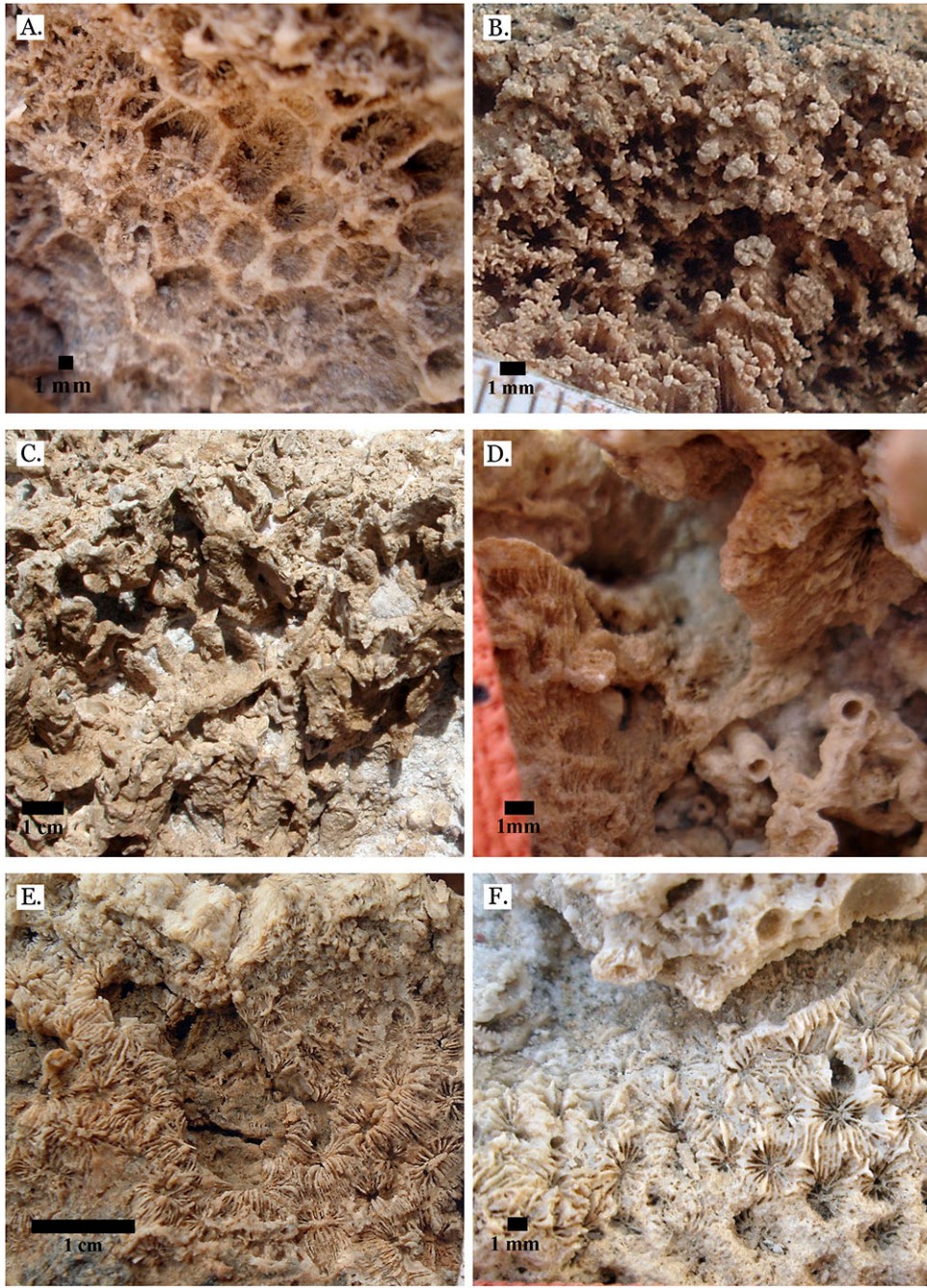

**Figure 18 Family Agariciidae.** (A) *Gardineroseris planulata* from Middle Pleistocene Wadi Wizr; (B) *Pavona* c.f. *bipartita* from Middle Pleistocene Wadi Gawasis; (C) *Pavona cactus* from Late Pleistocene Sharm Al Arab; (D) *Pavona cactus* from Late Pleistocene Sharm Al Arab; (E) *Pavona decussata* from Middle Pleistocene Wadi Gawasis; (F) *Pavona maldivensis* from Late Pleistocene Sharm Al Arab.

*lamellosa*, *Erythrastrea flabellate*, *Favites abdita*, *Leptastrea bottae*, *Lobophyllia hemprichii*, *Pavona minuta*, *Platygyra acuta*, and *Platygyra sinensis* all accounted for less than 1% of coral identifications (Table 3). The upper terrace at Wadi Wizr had the fourth highest H (1.92) and the third lowest Simpson's diversity index (0.718).

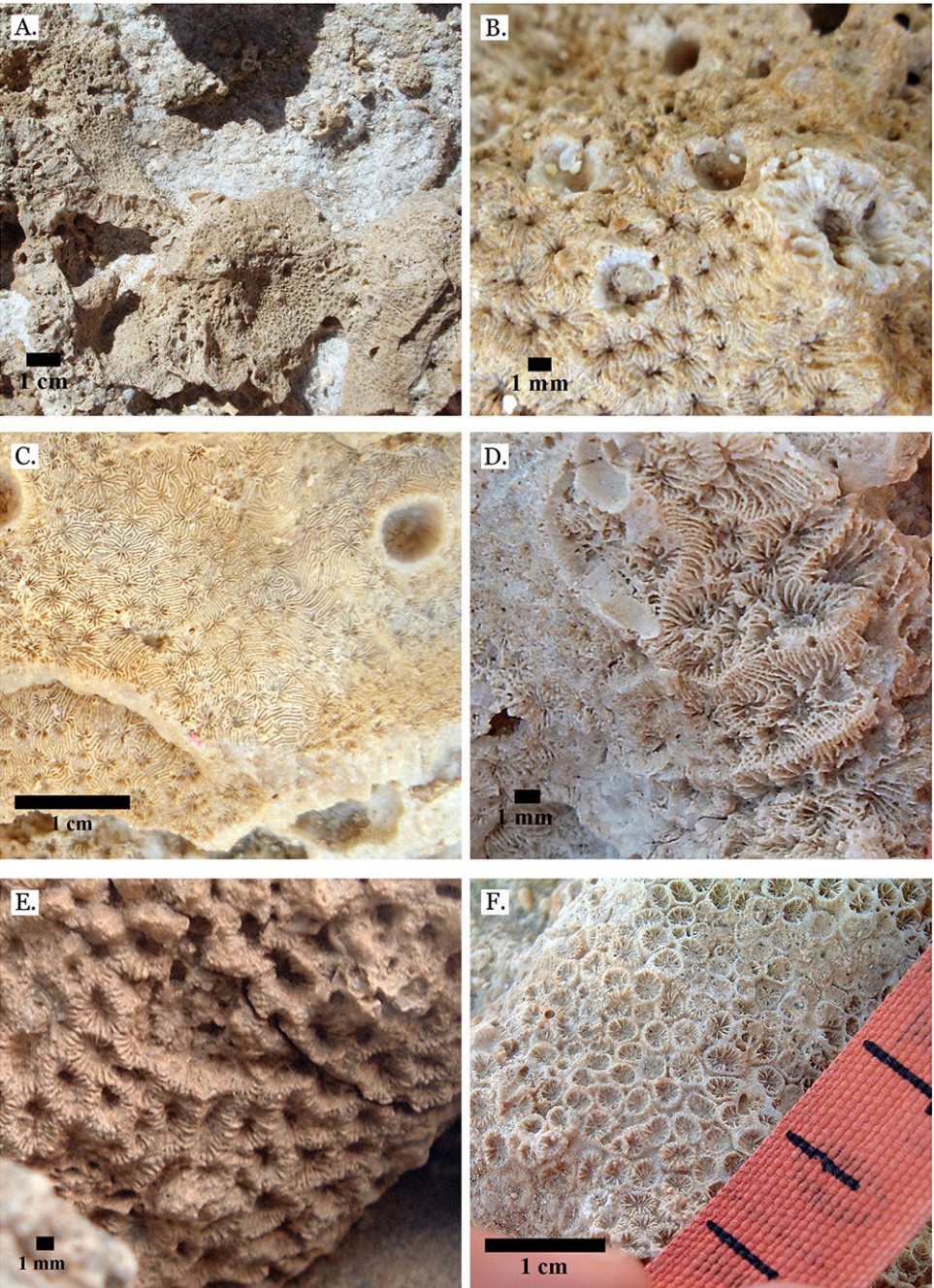

**Figure 19 Family Agariciidae.** (A) *Pavona frondifera* from Late Pleistocene Wadi Gawasis; (B) *Pavona frondifera* from Late Pleistocene Wadi Gawasis; (C) *Pavona minuta* from Wadi Wizr, Late Pleistocene or Holocene; (D) *Pavona venosa* from Late Pleistocene Sharm Al Arab. Family Siderastreidae. (E) *Siderastrea savignyana* from Middle Pleistocene Wadi Gawasis. Incertae sedis. (F) *Leptastrea pruinosa* from Late Pleistocene Wadi Gawasis.

### Wadi Wizr—lower terrace

I made 142 identifications on the Wadi Wizr Late Pleistocene lower terrace transect, including 11 coral species. Coral cover was 82.4%, coralline algae was 12%, and marine sand was 5.6% (Table 2). The most abundant coral were a mix of massive and

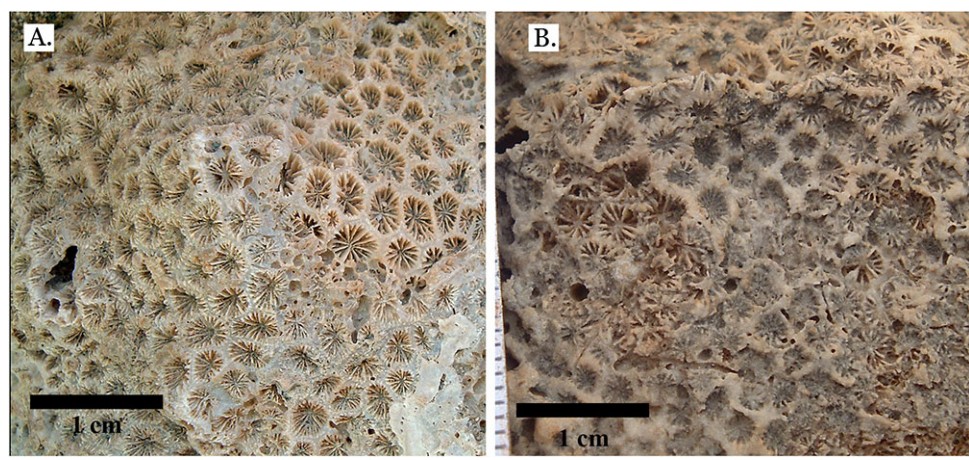

**Figure 20 Incertae sedis.** (A) *Leptastrea bottae* from Late Pleistocene Sharm Al Arab; (B) *Leptastrea bottae* from Middle Pleistocene Wadi Gawasis.

**Table 2 Summary of transect data from Middle and Late Pleistocene terraces.**

| Terrace | Total number of data points | Sediment type(s) | Sediment cover (% of total) | Coralline algae (% of total) | Coral cover (% of total) | Number of coral taxa | Shannon–Wiener index | Simpson |
|---|---|---|---|---|---|---|---|---|
| **Middle Pleistocene** | | | | | | | | |
| *Wadi Wizr* | 130 | Carbonate mud | 8.5 | 12.3 | 60 | 9 | 1.57 | 0.732 |
| | | Marine sand | 19.2 | | | | | |
| *Wadi Gawasis* | 179 | Siliclastic | 12.9 | 0 | 78 | 24 | 2.42 | 0.844 |
| | | Marine sand | 9 | | | | | |
| *Sharm Al Arab* | 110 | Marine sand | 28.2 | 3.6 | 68.2 | 7 | 1.16 | 0.569 |
| **Late Pleistocene** | | | | | | | | |
| *Wadi Wizr: upper* | 129 | Marine sand | 1.6 | 0.8 | 97.7 | 20 | 1.92 | 0.718 |
| *Wadi Wizr: lower* | 142 | Marine sand | 5.6 | 12 | 82.4 | 11 | 1.75 | 0.728 |
| *Wadi Gawasis: upper* | 135 | Marine sand | 4.4 | 0.7 | 94.8 | 16 | 1.74 | 0.634 |
| *Wadi Gawasis: lower* | 146 | Marine sand | 1.4 | 4.8 | 93.8 | 28 | 2.95 | 0.919 |
| *Sharm Al Arab: lower* | 158 | Marine sand | 10.2 | 3.8 | 86 | 30 | 2.87 | 0.915 |

branched *Porites* spp. (57.3% of coral identifications). *Galaxea fascicularis* (18.8%) was the second most abundant, followed by *Pocillopora damicornis* (7.7%). *Goniastrea stelligera* and *Favites pentagona* were 4.3% each, and *Platygyra daedalea* was 2.6%. *Lobophyllia hemprichii*, *Cyphastrea microphthalma*, *Echinopora gemmacea*, *Dipsastraea speciosa*, and *Favites paraflexuosus* all accounted for 0.9% of coral identifications (Table 3). The lower Wadi Wizr terrace had the fourth lowest H and Simpson's diversity index (1.75 and 0.728, respectively).

### Wadi Gawasis—upper terrace

One hundred thirty-five identifications of taxa or sediment type were made on the Wadi Gawasis Late Pleistocene upper terrace transect, including 16 species. Coral cover

**Table 3** Species abundances for all taxa identified from Middle and Late Pleistocene terraces.

| | |
|---|---|
| **Wadi Wizr, Middle Pleistocene** | |
| *Porites* spp. | 65.4% |
| *Goniastrea stelligera* | 20.5 |
| *Stylocoeniella guentheri* | 6.4 |
| *Cyphastrea microphthalma* | 1.28 |
| *Gardineroseris planulata* | 1.28 |
| *Dipsastraea* sp. | 1.28 |
| *Goniastrea* sp. | 1.28 |
| *Astrea curta* | 1.28 |
| *Platygyra sinensis* | 1.28 |
| **Wadi Gawasis, Middle Pleistocene** | |
| *Porites* spp. | 42.9% |
| *Galaxea fascicularis* | 9.3 |
| *Echinopora lamellosa* | 8.6 |
| *Favites micropentagonus* | 7.1 |
| *Goniopora* sp. | 7.1 |
| *Pocillopora damicornis* | 2.9 |
| *Stylocoeniella guentheri* | 2.9 |
| Faviid sp. | 2.9 |
| *Goniastrea retiformis* | 2.1 |
| *Astrea curta* | 2.1 |
| *Goniastrea stelligera* | 1.4 |
| *Leptastrea bottae* | 1.4 |
| *Pavona* c.f. *bipartite* | 1.4 |
| *Acropora* sp. | 0.7 |
| *Cyphastrea seralia* | 0.7 |
| *Dipsastrea pallida* | 0.7 |
| *Dipsastraea speciosa* | 0.7 |
| *Echinopora gemmacea* | 0.7 |
| *Gardineroseris planulata* | 0.7 |
| *Gyrosmilia interrupta* | 0.7 |
| *Pavona decussate* | 0.7 |
| *Siderastrea savignyana* | 0.7 |
| *Favites* sp. | 0.7 |
| *Goniastrea* sp. | 0.7 |

(*Continued*)

| Table 3 (continued). | |
|---|---|
| **Sharm Al Arab, Middle Pleistocene** | |
| *Porites* spp. | 86.7% |
| Agariciid sp. | 2.7 |
| *Echinopora lamellosa* | 2.7 |
| *Astrea curta* | 2.7 |
| *Stylocoeniella guentheri* | 2.7 |
| *Leptoria phrygia* | 1.3 |
| *Pocillopora damicornis* | 1.3 |
| **Wadi Wizr, Late Pleistocene, lower** | |
| *Porites* spp. | 57.3% |
| *Galaxea fascicularis* | 18.8 |
| *Pocillopora damicornis* | 7.7 |
| *Goniastrea stelligera* | 4.3 |
| *Favites pentagona* | 4.3 |
| *Platygyra daedalea* | 2.6 |
| *Lobophyllia hemprichii* | 1.7 |
| *Cyphastrea microphthalma* | 0.9 |
| *Dipsastraea speciosa* | 0.9 |
| *Echinopora gemmacea* | 0.9 |
| *Favites paraflexuosus* | 0.9 |
| **Wadi Wizr, Late Pleistocene, upper** | |
| *Porites* spp. | 50% |
| *Galaxea fascicularis* | 19 |
| *Pocillopora damicornis* | 5.6 |
| *Dipsastraea speciosa* | 3.2 |
| *Goniastrea stelligera* | 3.2 |
| *Favites pentagona* | 3.2 |
| *Acropora* spp. | 2.4 |
| *Dipsastraea pallida* | 2.4 |
| *Favites vasta* | 1.6 |
| *Hydnophora microconos* | 1.6 |
| *Dipsastraea favus* | 0.8 |
| *Echinopora gemmacea* | 0.8 |
| *Echinopora lamellosa* | 0.8 |
| *Erythrastrea flabellate* | 0.8 |
| *Favites abdita* | 0.8 |
| *Leptastrea bottae* | 0.8 |

| Table 3 (continued). | |
|---|---|
| *Lobophyllia hemprichii* | 0.8 |
| *Pavona minuta* | 0.8 |
| *Platygyra acuta* | 0.8 |
| *Platygyra sinensis* | 0.8 |
| **Wadi Gawasis, Late Pleistocene, lower** | |
| *Porites* spp. | 22.6% |
| *Galaxea fascicularis* | 13.2 |
| *Goniastrea stelligera* | 6.6 |
| *Platygyra daedalea* | 6.6 |
| *Pocillopora damicornis* | 5.1 |
| *Leptastrea bottae* | 5.1 |
| *Dipsastraea speciosa* | 4.4 |
| *Lobophyllia hemprichii* | 3.6 |
| *Echinopora lamellose* | 3.7 |
| *Acanthastrea echinata* | 2.9 |
| *Favites pentagona* | 2.9 |
| *Goniastrea pectinata* | 2.9 |
| *Favites paraflexuosus* | 2.2 |
| *Hydnophora microconos* | 2.2 |
| *Leptoria Phrygia* | 2.2 |
| *Platygyra acuta* | 2.2 |
| *Cyphastrea microphthalma* | 1.5 |
| *Dipsastrea favus* | 1.5 |
| *Favites flexuosa* | 1.5 |
| *Platygyra lamellina* | 1.5 |
| *Acanthastrea rotundoflora* | 0.7 |
| *Dipsastrea matthaii* | 0.7 |
| *Favites abdita* | 0.7 |
| *Favites vasta* | 0.7 |
| *Leptastrea pruinosa* | 0.7 |
| *Pavona frondifera* | 0.7 |
| *Platygyra sinensis* | 0.7 |
| *Acropora* sp. | 0.7 |
| **Wadi Gawasis, Late Pleistocene, upper** | |
| *Porites* spp. | 64% |
| *Pocillopora damicornis* | 7.2 |
| *Platygyra acuta* | 5.6 |

(Continued)

| Table 3 (continued). | |
|---|---|
| *Platygyra daedalea* | 4 |
| *Dipsastraea speciosa* | 3.2 |
| *Leptoria Phrygia* | 3.2 |
| *Goniastrea pectinata* | 2.4 |
| *Cyphastrea microphthalma* | 1.6 |
| *Echinopora gemmacea* | 1.6 |
| *Favites pentagona* | 1.6 |
| *Leptastrea bottae* | 1.6 |
| *Acropora* sp. | 0.8 |
| *Favites paraflexuosus* | 0.8 |
| *Galaxea fascicularis* | 0.8 |
| *Hydnophora microconos* | 0.8 |
| *Platygyra lamellina* | 0.8 |
| **Sharm Al Arab, Late Pleistocene, lower** | |
| *Galaxea fascicularis* | 22.1% |
| *Porites* spp. | 15.4 |
| *Favites pentagona* | 7.4 |
| *Echinopora gemmacea* | 6.6 |
| *Pocillopora damicornis* | 6.6 |
| *Acropora* sp. | 5.1 |
| *Cyphastrea microphthalma* | 4.4 |
| *Dipsastrea pallida* | 3.7 |
| *Platygyra daedalea* | 3.7 |
| *Dipsastraea speciosa* | 2.2 |
| *Goniastrea stelligera* | 2.2 |
| *Goniastrea pectinata* | 2.2 |
| *Dipsastrea favus* | 1.5 |
| *Favites abdita* | 1.5 |
| *Favites flexuosa* | 1.5 |
| *Favites paraflexuosus* | 1.5 |
| *Leptastrea bottae* | 1.5 |
| *Pavona cactus* | 1.5 |
| *Platygyra acuta* | 1.5 |
| *Acanthastrea echinata* | 0.7 |
| *Caulastrea tumida* | 0.7 |
| *Cyphastrea seralia* | 0.7 |
| *Gyrosmilia interrupta* | 0.7 |

| Table 3 (continued). | |
|---|---|
| *Hydnophora microconos* | 0.7 |
| *Leptastrea pruinosa* | 0.7 |
| *Leptoria Phrygia* | 0.7 |
| *Lobophyllia hemprichii* | 0.7 |
| *Pavona maldivensis* | 0.7 |
| *Pavona venosa* | 0.7 |
| *Platygyra sinensis* | 0.7 |

was 94.8%, marine sand was 4.4%, and coralline algae was 0.7% (Table 2). The most abundant coral were primarily branched *Porites* spp. (64% of coral identifications). *Pocillopora damicornis* (7.2%) was the second most abundant, followed by *Platygyra acuta* (5.6%) and *Platygyra daedalea* (4%). *Dipsastraea speciosa* and *Leptoria phrygia* were 3.2% each, *Goniastrea pectinate* was 2.4%, and *Cyphastrea microphthalma*, *Echinopora gemmacea*, *Favites pentagona*, and *Leptastrea bottae* were 1.6% each. *Favites paraflexuosus*, *Galaxea fascicularis*, *Hydnophora microconos*, *Platygyra lamellina*, and *Acropora* sp. all accounted for less than 1% of coral identifications (Table 3). The upper terrace of Wadi Gawasis had the third lowest H (1.74) and the second lowest Simpson's diversity index (0.634).

### Wadi Gawasis—lower terrace

I made 146 identifications on the Wadi Gawasis Late Pleistocene lower terrace transect, including 28 species. Coral cover was 93.8%, coralline algae was 4.8%, and marine sand was 1.4% (Table 2). The most abundant coral were primarily branched *Porites* spp. (22.6% of coral identifications). *Galaxea fascicularis* (13.2%) was the second most abundant, followed by *Platygyra daedalea* and *Goniastrea stelligera* at 6.6% each. *Leptastrea bottae* and *Pocillopora damicornis* were 5.1% each, followed by *Dipsastraea speciosa* (4.4%), *Lobophyllia hemprichii* (3.6%), and *Echinopora lamellosa* (3.7%). *Acanthastrea echinata*, *Favites pentagona*, and *Goniastrea pectinata* were 2.9% each, and *Favites paraflexuosus*, *Hydnophora microconos*, *Leptoria phrygia*, and *Platygyra acuta* were all 2.2% each. *Cyphastrea microphthalma*, *Dipsastraea favus*, *Favites flexuosa*, and *Platygyra lamellina* were 1.5% each while *Acanthastrea rotundoflora*, *Dipsastraea matthaii*, *Favites abdita*, *Favites vasta*, *Leptastrea pruinosa*, *Pavona frondifera*, *Platygyra sinensis*, and *Acropora* sp. all accounted for less than 1% of coral identifications (Table 3). The lower terrace at Wadi Gawasis had the highest diversity of all the terraces with an H of 2.95 and a Simpson's diversity index of 0.919.

### Sharm el Arab—lower terrace

One hundred fifty-eight identifications were made on the Wadi Gawasis Late Pleistocene lower terrace transect, including 31 species. Coral cover was 86%, marine sand was 10.2%, and coralline algae was 3.8%. The most abundant coral was *Galaxea fascicularis* (22.1% of coral identifications). The second most abundant were primarily branching *Porites* spp.

(15.4%), followed by *Favites pentagona* (7.4%), *Echinopora gemmacea*, and *Pocillopora damicornis* at 6% each, and *Acropora* sp. at 5.1%. *Cyphastrea microphthalma* was 4%, *Dipsastraea pallida* and *Platygyra daedalea* were 3.7% each, and *Dipsastraea speciosa*, *Goniastrea stelligera*, and *Goniastrea pectinata* were 2.2% each. *Dipsastraea favus*, *Favites abdita*, *Favites flexuosa*, *Favites paraflexuosus*, *Leptastrea bottae*, *Pavona cactus*, and *Platygyra acuta* were each 1.5%, while *Acanthastrea echinata*, *Caulastrea tumida*, *Cyphastrea seralia*, *Gyrosmilia interrupta*, *Hydnophora microconos*, *Leptastrea pruinosa*, *Leptoria phrygia*, *Lobophyllia hemprichii*, *Platygyra sinensis*, *Pavona maldivensis*, and *Pavona venosa* all accounted for less than 1% of coral identifications. The lower terrace at Sharm Al Arab had the second highest diversity of all the terraces, with an H of 2.87 and a Simpson's diversity index of 0.915.

## DISCUSSION

### Pleistocene diversity and paleoenvironments

A common feature of coral reefs is the change in species and coral growth forms with physical parameters such as available light, sediment input, subaerial exposure, and hydrodynamics. For the most part these physical parameters vary with depth, so changes in species composition often correlate with depth (*Perrin, Bosence & Rosen, 1995*; *Chappell, 1980*). Zonation can be characterized by framework density (percentage of coral and coralline algae cover), diversity, and the relative abundance of coral species (*Perrin, Bosence & Rosen, 1995*; *Montaggioni, 2005*).

Although local conditions determine zonation on specific reefs, there are general trends that usually apply (*Chappell, 1980*). For example, species richness is lowest on reef flats. It increases with depth, then decreases again as wave energy increases near the reef edge. Richness increases again with depth on the fore-reef slope as wave energy decreases, then decreases again with increasing depth as light availability decreases. Back reefs have the greatest sedimentation, and intermediate diversity. Again, diversity increases with depth on back reefs, then decreases again as available light decreases (*Chappell, 1980*). Because zonation is preserved on fossil reefs, assemblage data can be used to infer their paleoenvironments (*Mesolella, 1967*; *Edinger, Pandolfi & Kelley, 2001*), even in the absence of reliable paleodepths. Although one should be cautious about assuming that environmental preferences of organisms remain unchanged through time (*Bottjer & Jablonski, 1988*), Pleistocene reef environments show similar zonation trends to modern reefs (*Mesolella, 1967*; *Jackson, 1992*; *Pandolfi, 1996*; *Pandolfi & Jackson, 2006*; *Alexandroff, Zuschin & Kroh, 2016*).

### *Middle Pleistocene terraces*

The Middle Pleistocene terraces at Wadi Wizr and Sharm Al Arab have the lowest species richness (Fig. 21), lowest coral cover, and highest sedimentation of all the examined Pleistocene terraces (Table 2). Low diversity and high sedimentation indicates a reef flat or shallow back reef zone (*Chappell, 1980*). *Loya (1972)* recorded an average of 8.37 (±3.27) species on 108 m transects on *Stylophora pistillata*-dominated back reef/reef flats on fringing reefs at Eilat in the Red Sea; Sharm Al Arab is within one standard deviation

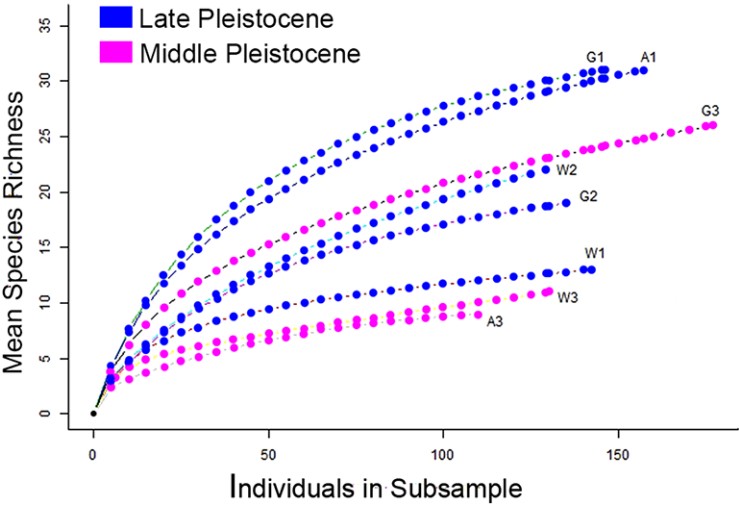

**Figure 21 Rarefaction curves showing the mean richness of Pleistocene terraces based on the number of sampled corals.** W1 = Late Pleistocene lower terrace at Wadi Wizr, W2 = Late Pleistocene upper terrace at Wadi Wizr, W3 = Middle Pleistocene terrace at Wadi Wizr, G1 = Late Pleistocene lower terrace at Wadi Gawasis, G2 = Late Pleistocene upper terrace at Wadi Gawasis, G3 = Middle Pleistocene terrace at Wadi Gawasis, A1 = Late Pleistocene lower terrace at Sharm Al Arab, A3 = Middle Pleistocene terrace at Sharm Al Arab.

of this average with seven species found on a 110 m transect (Table 2). Likewise Wadi Wizr is within one standard deviation of 9.37 (±3.49) species found on 135 m transects of back reef/reef flat (see Table 2 in *Loya, 1972*), with nine species found on a 130 m transect (Table 2). *Riegl & Velimirov (1994)* characterized coral communities of offshore and onshore fringing reefs with varying degrees of hydrodynamic exposure near Hurghada, Egypt. They did not specifically characterize back reef or reef flat zones, however they found *Porites* species only accounted for more than 40% of transect intercepts in sheltered reef environments (*Riegl & Velimirov, 1994*). Like nearly all of the terraces examined here, Middle Pleistocene Wadi Wizr and Sharm Al Arab are dominated by *Porites* (Table 3), which may indicate a sheltered environment. The presence of fine-grained carbonate mud at Wadi Wizr further supports the idea that it was a low energy, sheltered environment, because very fine-grained sediments only accumulate in low-energy environments.

*Riegl & Pillar (1999)* described different coral frameworks from the Red Sea. They reported high abundances of massive *Porites* from sheltered reef edges and reef slopes (68% ± 30% and 85% ± 29%, respectively), however reef edges and slopes do not usually have large amounts of sediment because they have higher wave action (*Chappell, 1980*). *Riegl & Pillar (1999)* also describe *Porites* "carpets," however these occur in environments without a pronounced slope and are dominated by columnar forms; Wadi Wizr and Sharm Al Arab are fringing reefs, which are sloped, and the *Porites* at both locations are usually massive (see Figs. 4A and 4B).

The most extensive analysis of coral communities of the modern Red Sea was provided by *Sheppard & Sheppard (1991)*. They surveyed 200 sites, estimating percent coral cover of all species by eye. As in this study, *Sheppard & Sheppard (1991)* found similar species of *Acropora* and *Porites* (as well as *Montipora*, *Goniopora*, and *Alveopora*) difficult to

distinguish, so they treated them as single taxon in ecological analyses. Using cluster analysis, they identified 13 distinct types of coral community located along the east coast of the Red Sea. These community descriptions include key features, such as dominant and characteristic taxa, degree of sedimentation, and hydrodynamic exposure. These features can be compared to data from this study to see if the same communities are found on Pleistocene terraces.

Of the communities described by *Sheppard & Sheppard (1991)*, Wadi Wizr and Sharm Al Arab most closely resemble community 4. It is dominated by *Porites lutea* (47%–85% cover) and is described thus:

> The very sheltered bays and back reef slopes of patch reefs in the north and central regions support this community of high *Porites* cover. Cover values provided by the enormous colonies of Porites range up to 85%, amongst which are the equally characteristic *Pavona variens*, *Pavona cactus*, the Pocilloporidae and soft coral genus *Xenia*. These habitats are always fairly sedimented.

Middle Pleistocene Wadi Wizr and Sharm Al Arab are dominated by *Porites* (65.4% and 86.7%, respectively) with a co-occurrence of *Pocillopora damicornis*; the main difference is that *Pavona* are absent from both sites.

The low species richness, dominance of massive *Porites*, and high percentage of sediment cover all point to a shallow back reef or reef flat environment for the Middle Pleistocene terraces at Wadi Wizr and Sharm Al Arab. The presence of fine-grained sediments at Wadi Wizr supports sheltered, low-energy conditions, as does the high occurrence of rhodolithic algae (see Fig. 3A). Likewise, large colonies of *Goniastrea stelligera* account for 20.5% of taxa at Wadi Wizr, and in the Red Sea they are only found in shallow, sheltered environments (*Sheppard & Sheppard, 1991*). Sharm Al Arab may have been a more exposed site, but the presence of *Leptoria Phrygia*, which is rarely found deeper than 4 m in the modern Red Sea (*Sheppard & Sheppard, 1991*), supports a shallow reef environment.

The Middle Pleistocene terrace at Wadi Gawasis has the highest diversity of all the Middle Pleistocene terraces, and the third highest species richness of all terraces (Fig. 21). It also has the third highest sediment cover; more than 12% of it non-marine sand deposited from land (Table 2). High sedimentation is typical of back reef slopes (*Chappell, 1980*), however this could also be a shallow fore-reef slope on a fringing reef with a very narrow back reef and reef flat. *Loya (1972)* found an average of 26.67% ± 3.87 species on 90 m transects on fore-reef slopes between 20 and 30 m, which is a similar to the 25 species found on 140 m in this study. This paleodepth could be feasible if *Lambeck et al.'s (2011)* hypothesis of tectonic uplift in the northern and central Red Sea is correct, but the 9.3% cover of *Galaxea fascicularis* may indicate Wadi Gawasis was at the shallower end of this range, as the modern depth range of *Galaxea fascicularis* on clear-water reef slopes is only 5–20 m (*Sheppard & Sheppard, 1991*). Fossil reefs also have higher diversity than living reefs because they are a combination of the living and death assemblages (*Edinger, Pandolfi & Kelley, 2001*), so Wadi Gawasis may actually

represent a reef slope that had lower live diversity, perhaps indicating a depth shallower than 20–30 m.

A mix of branching and massive forms of *Porites* spp. account for 42.9% of identified coral on the Middle Pleistocene Wadi Gawasis terrace (21.2% of total). *Riegl & Velimirov (1994)* only found *Porites* cover in this range on sheltered platform reefs at 12 m, and none of the reef environments they surveyed had such high species richness (see Figs. 6 and 8 in *Riegl & Velimirov, 1994*). The *Porites* reef framework described by *Riegl & Pillar (1999)* is dominated by massive *Porites* in shallow water and dominated by branching forms at depth, though they do not report an exact depth for the transition. Although not dominant, *Echinopora lamellosa* (8.6% of taxa) is the third most abundant species after *Porites* and *Galaxea fascicularis*. In the modern Red Sea this species is common on reef slopes deeper than 15 m (*Sheppard & Sheppard, 1991*).

The high species richness at Wadi Gawasis points to a back or fore-reef slope, and the modern depth range of *Galaxea fascicularis*, *Echinopora lamellosa*, and the mix of massive and branching *Porites* seem to indicate a depth range between 15 and 20 m.

### Late Pleistocene terraces

The Late Pleistocene lower terraces at Wadi Gawasis and Sharm Al Arab have the highest species richness of all the surveyed terraces (Fig. 21). Their very high diversity indicates a fore-reef slope environment according to *Chappell (1980)*. The highest diversity found by *Loya (1972)* is an average of 26.67 (±3.87) species on 90 m transects for reef slopes at a depth of 20–30 m. Wadi Gawasis had 28 species on a 146 m transect and Sharm Al Arab had 30 on 158 m. Although both terraces have a high occurrence of *Porites* (22.6% at Wadi Gawasis and 15.4% of identified corals at Sharm Al Arab) neither has the very high percent cover seen on sheltered reefs by *Riegl & Velimirov (1994)*. Exposed and semi-exposed reefs surveyed by the same authors are dominated by *Acropora* and *Millipora*, respectively, and these species only occur in very low percentages, if at all, on the lower terraces at Wadi Gawasis and Sharm Al Arab (Table 3). Likewise, these terraces do not clearly fit into any of the defined coral carpet or framework reef types described from the modern Red Sea by *Riegl & Pillar (1999)*, although they are closest to the *Porites* framework reef for which they report a *Porites* cover of 67% ± 30%.

These two terraces do bear some resemblance to community 5 described thus by *Sheppard & Sheppard (1991)*:

> A widespread community of high diversity is found in the north and central region of the Red Sea where there is moderate exposure. They do not differ greatly from community 3 [found between 5 and 15 m], except that they may be dominated by one or two species. Three sub-groups appear according to whether one of the component stony corals attains dominance or not: the most common is one dominated by the hydrozoan *Millepora*, another small group is dominated by *Goniopora* . . . while the third group lacks any strongly dominant species.

*Porites* are the most abundant taxa at Wadi Gawasis (22.6%) followed by *Galaxea fascicularis* (11.7%), while *Galaxea fascicularis* accounts for 22.1% of all taxa at Sharm

Al Arab, and *Porites* 15.4%. Although neither of these taxa are identified by Sheppard and Sheppard as being one of the three sub-groups, these terraces do have high diversity, and they are "dominated by one or two species." A shallow slope environment is also supported by the relatively high abundance of shallow water species (occurring shallower than 25 m) on both terraces. These include *Goniastrea stelligera*, *Pocillopora damicornis*, *Lobophyllia hemprichii*, *Acanthastrea echinata*, and *Hydnophora microconos* (see Table 2, *Sheppard & Sheppard, 1991*). *Galaxea fascicularis* is not reported as a dominant or even common taxon on most modern Red Sea reefs (*Heiss et al., 2005*; *Riegl & Velimirov, 1994*; *Riegl & Piller, 2000*; *Sheppard & Sheppard, 1991*), however it is reported on some slopes between 5 and 20 m, and as very abundant (up to 50% cover) in very shallow, turbid environments (*Sheppard & Sheppard, 1991*). It could be that's its relatively high abundance on the Wadi Gawasis and Sharm Al Arab terraces indicates a turbid environment, or it may be that this taxon was more common on Pleistocene reefs than it is today.

The case for a relatively shallow fore-reef slope is strengthened by the poorly sorted coarse-grained marine sand found on both terraces, which is typical on reef slopes below the reef crest where rubble dominates sediments, and the deep fore-slope, where sediments are usually much finer (*Dudley, 1996*). The high occurrence of marine sand (10.2% of transect points) at Sharm Al Arab also strengthens the argument for a relatively turbid environment, because the finer grains in this poorly sorted sediment would have been available for resuspension if hydrodynamic conditions allowed. The very high diversity, high coral cover, and abundance of shallow water taxa indicate the Late Pleistocene lower terraces at Wadi Gawasis and Sharm Al Arab represent the shallow fore-reef slopes of fringing reefs. Comparisons to *Sheppard & Sheppard's (1991)* community 5 suggest it may be a moderately exposed site and shallower than 15 m. The high abundance of *Galaxea fascicularis* may mean it was also relatively turbid.

The upper and lower Late Pleistocene terraces along the Red Sea coast have both been dated to the MIS 5e, and the platform morphology was determined to be a result of erosion (*Plaziat et al., 2008*), therefore the upper terrace and lower terrace represent two depths on the same fringing reef. The upper terrace is approximately 2.5 m higher (shallower) than the lower terrace. At Wadi Gawasis, the upper terrace has lower species richness than the lower terrace, which meets the expectation of increasing diversity with depth on shallow fore-reef slopes (*Chappell, 1980*). Sixteen species were identified from 135 m on the upper terrace at Wadi Gawasis, which is within range of the 12.33 ± 4.89 species reported from 120 m transects at depths of 3–13 m by *Loya (1972).*

Branching forms of *Porites* spp. account for 64% of taxa on the upper terrace. This abundance is consistent with reef edge or shallow reef slope environments on sheltered reefs reported by *Riegl & Pillar (1999)*, however they note that massive forms of *Porites* dominate at the reef edge and upper slope, and columnar forms only become more common at depth. The general trend in coral growth form (independent of taxon) is that growth becomes more massive with increasing hydrodynamic stress, but more branching with increasing sedimentation (*Chappell, 1980*; *Perrin, Bosence & Rosen, 1995*). If the Wadi Gawasis site was relatively turbid, it may account for the branching growth form of *Porites* at shallower than expected depths.

Based on the high abundance of *Porites*, the upper terrace at Wadi Gawasis most closely resembles community 4 described by *Sheppard & Sheppard (1991)*, however the terrace lacks the high sediment cover characteristic of this community (Table 2). Even without an exact corollary in the literature on modern reefs of the Red Sea, the moderate diversity and proximity to the lower terrace strongly suggests this was a shallow upper slope environment.

The Late Pleistocene lower terrace at Wadi Wizr has the lowest diversity of all the Late Pleistocene terraces (Fig. 21). This is consistent with a reef flat or shallow back reef zone (*Chappell, 1980*). *Loya (1972)* found an average of 9.37 (±3.49) species on 135 m transects in back reef and reef flat environments, and the lower terrace at Wadi Wizr had 11 species on 142 m transects, which is within one standard deviation of Loya's average.

A mix of massive and branched *Porites* accounted for 57.3% of taxa, which is near to the values found on sheltered platform and fringing reefs by *Riegl & Velimirov (1994)*, and within range of sheltered reef edges (68% ± 30%), and very nearly in range of reef slopes (85% ± 29%) found by *Riegl & Pillar (1999)* on *Porites* framework reefs. This terrace doesn't fit well with any communities described by *Sheppard & Sheppard (1991)*, however their analysis excluded reef flats.

Theoretically, diversity on reef flats first increases from reef crest toward the back reef as wave energy decreases, then begins to decrease again as subaerial exposure increases (*Chappell, 1980*). In this framework, the lower terrace at Wadi Wizr could be a reef flat close to the reef edge, or a reef edge. This is further supported by the relatively high abundance of coralline algae (12%), which is characteristic of reef crests (edges) (*Perrin, Bosence & Rosen, 1995*; *Riegl & Pillar, 1999*).

In this interpretation the higher diversity of the Late Pleistocene upper terrace at Wadi Wizr can be explained as the higher diversity expected on mid reef flats relative to reef edges, where wave energy is minimal and subaerial exposure is not yet a factor (*Chappell, 1980*). The high abundance of shallow water taxa on both terraces, including *Pocillopora damicornis*, *Goniastrea stelligera*, and *Lobophyllia hemprichii* supports the idea of a shallow water environment (Table 3), and the high abundance of *Galaxea fascicularis* on both the lower (18.8%) and upper (19%) terraces (Table 3) suggests a turbid environment, which is consistent with a wave-washed reef flat. The upper terrace at Wadi Wizr is dominated by branching Porites (50%), which may also be expected in a turbid environment, and the mix of branching and massive *Porites* on the lower terrace might be expected on a turbid reef edge.

## Comparing coral assemblages

Data from this study is insufficient for a rigorous statistical analysis of community change across Pleistocene glaciation events (*Pandolfi, 1996*; *Pandolfi & Jackson, 2006*), but a qualitative analysis can still reveal which modern taxa are also present in the Middle and Late Pleistocene, if species may have gone locally extinct, and when endemics appeared.

### Middle and Late Pleistocene assemblages

In this study, only 30.7% of identified taxa occurred on both Middle and Late Pleistocene terraces. A total of 26.9% were only found on Middle Pleistocene terraces and 42%

were only found on Late Pleistocene terraces. However, this is most likely an artifact of sampling bias: five transects were carried out on Late Pleistocene reefs, and only three were carried out on Middle Pleistocene reefs. This bias is further exacerbated by the paleoenvironments preserved, because two of the three Middle Pleistocene terraces preserve low-diversity, back reef environments. When species data are compiled for all the published studies on Red Sea fossil reefs (Table 1) only one species, *Favites micropentagonus*, is found exclusively on Middle Pleistocene terraces.

*Favites micropentagonus* was first described in 2000 (*Veron, 2000*). Originally recognized from locations within the Coral Triangle, a marine biodiversity hotspot centered on Indonesia and Papua New Guinea, it has since been identified on reefs in the Seychelles Archipelago in the western Indian Ocean (*Sheppard & Obura, 2005*), the Gulf of Aden (*DeVantier et al., 2004*), the Gulf of Oman, eastern Africa, Madagascar, Chagos Archipelago, Thailand, Southeast Asia, Vietnam, southern Japan, Papua New Guinea, southern coast of Solomon Islands, and Micronesia (*DeVantier et al., 2008*). Fossil specimens have been reported from the Miocene of Fiji, the Quaternary of Indonesia (*Bromfield, 2013*), and the Quaternary of Japan (*Humblet, Iryu & Nakamori, 2009*). The coral reefs of the Red Sea are understudied relative to those found in the Caribbean or on Australia's Great Barrier Reef (*Berumen et al., 2013*), so it's possible *Favites micropentagonus*'s current range extends into the Red Sea but it hasn't been recognized yet. It may also have lived there during the Middle Pleistocene, suffered local extinction, and failed to recolonize the Red Sea during later interglacial periods.

*Pavona decusassata* and *Stylocoeniella guentheri* are the only other species identified from the Middle Pleistocene in this study that have not been reported from the Late Pleistocene anywhere else in the Red Sea, although they are reported from the modern Red Sea. This could be another artifact of inadequate sampling of fossil reefs at all time periods in this region, however *Stylocoeniella guentheri* at least is reported as a very common coral in the modern Red Sea, occurring from 5 to 50 m, in every environment from lagoonal back reefs to deep fore slopes, in clear and turbid water, and in any degree of hydrodynamic exposure from sheltered to exposed (*Sheppard & Sheppard, 1991*). It's relatively high abundance on all three Middle Pleistocene terraces in this study (Table 3) suggests that it was equally common during the Middle Pleistocene. If it lived in the Red Sea during the Late Pleistocene, it seems highly probably that it would have been found already.

The individual-based rarefaction curves in Fig. 21 provide a reliable comparison of species richness between Middle and Late Pleistocene coral terraces despite differences in sampling intensity (*Gotelli & Colwell, 2001*). On living reefs, the expectation is that diversity varies with zonation (*Chappell, 1980*; *Perrin, Bosence & Rosen, 1995*). That's why in his study of fossil coral reefs in Papau New Guinea, *Pandolfi (1996)* only compared communities belonging to the same reef zone to see if diversity had change through time. In this study, there are six reef zones represented on eight reef terraces, so it isn't possible to perform a rigorous, statistical comparison of zone diversity across time. However, *Pandolfi's study (1996)* and a similar study by *Pandolfi & Jackson (2006)* in the Caribbean, both found diversity of zones stable across Pleistocene glaciation events. My results meet the expectation that diversity varies by reef zone, and there is no strong temporal

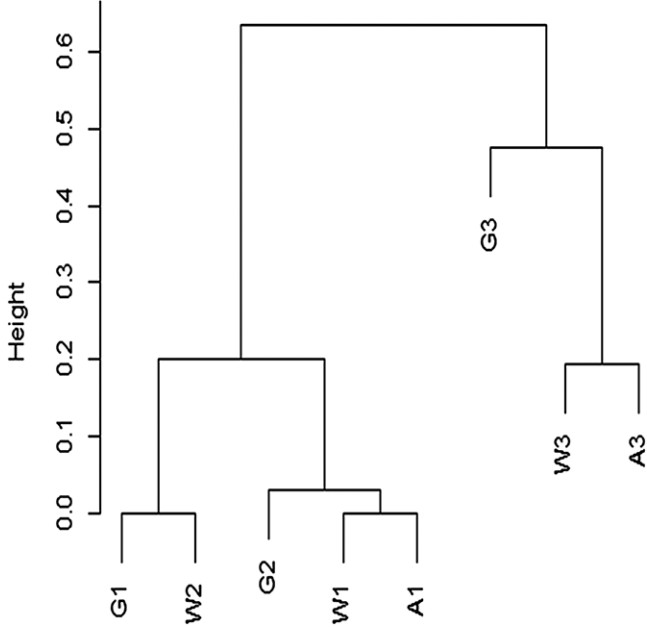

**Figure 22 Cluster analysis of Chao dissimilarity index.** W1 = Late Pleistocene lower terrace at Wadi Wizr, W2 = Late Pleistocene upper terrace at Wadi Wizr, W3 = Middle Pleistocene terrace at Wadi Wizr, G1 = Late Pleistocene lower terrace at Wadi Gawasis, G2 = Late Pleistocene upper terrace at Wadi Gawasis, G3 = Middle Pleistocene terrace at Wadi Gawasis, A1 = Late Pleistocene lower terrace at Sharm Al Arab, A3 = Middle Pleistocene terrace at Sharm Al Arab.

pattern that can't be explained by differences in zonation, which is consistent with the previous studies (*Pandolfi, 1996*; *Pandolfi & Jackson, 2006*).

The diversity found on each terrace is consistent with the theoretical expectations for each reef zone identified for it. For example, the lower terraces of the Late Pleistocene at Wadi Gawasis (G1) and Sharm Al Arab (A1) are both likely shallow fore-reef slopes; they have the highest diversity of all the studied terraces, which is consistent with fore-reef slopes being the most diverse zone on a fringing reef (*Chappell, 1980*). The Middle Pleistocene terraces at Wadi Wizr (W3) and Sharm Al Arab (A3) are both likely sheltered back reef environments, and they have the lowest diversity of all the terraces, which is expected for shallow back reefs (*Chappell, 1980*). Relative diversity on the remaining terraces is also consistent with the theoretical expectations for diversity in different reef zones. For example, the Late Pleistocene reef crest at Wadi Wizr (W1) has low diversity similar to the Middle Pleistocene back reefs (W3 and A3), while the Middle Pleistocene shallow slope (back or fore-reef) at Wadi Gawasis (G3) has moderately high diversity, similar to a reef flat with limited exposure and limited wave stress found on the upper terrace of Wadi Wizr (W2) of the Late Pleistocene.

Although species richness is determined by reef zone, a cluster analysis using the Chao dissimilarity index shows terraces grouping by age rather than environment (Fig. 22). When *Pandolfi (1996)* and *Pandolfi & Jackson (2006)* performed similar analysis, they found that reef communities clustered by geographical location rather than time period, indicating no significant change in reef species across Pleistocene glaciations.

The small number of transects used in this analysis means that any interpretation needs to be tested with more data, specifically with more terraces of the same reef zones. However, one simple explanation for the difference between my cluster results and those of *Pandolfi (1996)* is in our different approach to rare taxa. The Chao dissimilarity index weights shared rare taxa, which I included in my analysis, while *Pandolfi (1996)* excluded rare taxa from his analysis. It makes sense that rare taxa would be more vulnerable to environmental shifts, and therefore more likely to change through time. They are also less likely to be sampled, and therefore less reliable for characterizing communities.

### Pleistocene and modern assemblages

All but two taxa reported from the Pleistocene of the Red Sea are reported from the modern Red Sea: *Favites micropentagonus* (see Discussion), and *Pavona minuta*. *Pavona minuta* has been reported from the Late Pleistocene of the Red Sea by *Dullo (1990)*, and from the modern Gulf of Oman (*DiBattista et al., 2015*). It's possible that *Pavona minuta* occurs so rarely in the modern Red Sea that it has yet to be identified, or it may have gone locally extinct during the last glacial low stand. *Pavona minuta* also resembles *Pavona duerdeni*, a species that is found in the modern Red Sea but which has larger corallites and more exert septo-costae. Thus another possibility is that these Pleistocene *Pavona minuta* are morphological variants of *Pavona duerdeni*, or that *Pavona minuta* in the modern Red Sea are routinely misidentified as *Pavona duerdeni*. Based on preservation (see Fig. 19C) the one *Pavona minuta* specimen identified in this study looks younger than Late Pleistocene, and may actually be Holocene-aged.

*DiBattista et al. (2015)* report twenty-one endemic coral species from the modern Red Sea, and none of them have been reported from the Pleistocene of the Red Sea. Many Red Sea endemics have only been described recently: 13 of 21 have been described since 2000 (*DiBattista et al., 2015*). It is possible that they have been found by earlier fossil workers but misidentified; however, *DeVantier et al. (2000)* indicate most of these newly described taxa are distinctive and unlikely to be confused with another species. It is more likely that they haven't been found because (1) they are rare, and (2) Pleistocene Red Sea reefs are under-sampled. The recently described Red Sea endemics occur in relatively low abundances (*DeVantier et al., 2000*), and currently 121 coral species have been identified from the Pleistocene of the Red Sea (Table 1), while 346 have been reported from the Modern Red Sea (*DiBattista et al., 2015*). Even if taxa were significantly different between the Pleistocene and the present, there is no reason to think diversity would be lower in the past, which suggests two-thirds of Pleistocene coral species have yet to be identified.

Both *Erythrastrea flabellata* and *Favites vasta* were once thought to be endemic to the Red Sea (*DeVantier et al., 2000*), but recently they have also been reported from the Gulf of Aden and Socotra outside the Bab al Mandab (*DiBattista et al., 2015*). To the best of my knowledge, this study is the first report of these species from the Pleistocene: a single colony of *Erythrastrea flabellata* was found on the Late Pleistocene upper terrace at Wadi Wizr where *Favites vasta* was also found. The latter was also found on the Late

Pleistocene lower terrace at Wadi Gawasis. As more fossil terraces are studied it seems highly probably that more rare and endemic species will be found.

## Reef stability through time

Although some species may have gone locally extinct, the eight Egyptian reef terraces from the Middle Pleistocene share most of the same taxa as modern reefs, and have similar zonation as modern Red Sea reefs. This suggests that the coral-reef communities of the Red Sea have been relatively stable across glaciations for at least 250,000 years. The stability of coral reefs across Pleistocene glaciations has been documented elsewhere in the Indo-Pacific (*Pandolfi, 1996*) and Caribbean (*Jackson, 1992*; *Pandolfi & Jackson, 2006*); however, the result is more surprising for the Red Sea than for other locations. The changes to the physical environment during sea-level low stands made the Red Sea particularly adverse to life; either coral communities had to persist in extreme environments, or they went locally extinct and recolonized the entire basin after each glaciation.

During the Penultimate (MIS 6) and Last (MIS 2) Glacial Maximums, the Red Sea Basin's already narrow and shallow connection to the wider Indian Ocean became highly restricted, with depths as shallow as 17 m at the shallowest point, the Hanish Sill (*Fernandes, Rohling & Siddall, 2006*; *Siddall et al., 2004*). This resulted in net evaporation that caused salinities to increase dramatically, with average values over 50‰ in surface and bottom waters (*Almogi-Labin, 1982*; *Almogi-Labin, Hemleben & Meischner, 1998*; *Badawi, Schmiedl & Hemleben, 2005*; *Fenton et al., 2000*; *Hemleben et al., 1996*; *Thunnell, Locke & Williams, 1988*). Surface waters during the Last Glacial Maximum have been reconstructed to 57‰ in the north, 53‰ in the central Red Sea, and 47‰ in the south (*Fenton et al., 2000*; *Hemleben et al., 1996*), resulting in a complete disappearance of planktonic foraminifera (which have a salinity tolerance of 49‰) in the north and central Red Sea (*Hemleben et al., 1996*). *Badawi, Schmiedl & Hemleben (2005)* also found higher salinities in the north than in the south during glacial periods.

Stronger winds in the north resulted in increased dust input leading to increased flux of organic matter during MIS 2, 4, and 6 (*Badawi, Schmiedl & Hemleben, 2005*). But while higher surface productivity is characteristic of glacial oceans in general, the precipitous rise in salinity is an obstacle to survival during glaciations specific to the Red Sea. Based on turnover in reef-associated molluscan fauna between the Late Pleistocene and modern Red Sea, *Taviani (1998)* proposed a series of "complete biotic recolonizations" of all reef fauna from the Indian Ocean after mass extinction during glacial periods. Earlier authors had proposed a similar scenario of extreme salinities wiping out local fauna, requiring subsequent recolonizations when sea level rose again (*Gvirtzman et al., 1977*), and, more recently, it has been suggested specifically for corals (*Coles, 2003*; *Sheppard & Sheppard, 1991*).

An alternative hypothesis is that some corals persisted during glacial low stands in refugia, especially in the Gulf of Aqaba and the southern Red Sea. Genetic evidence provides the best support for this hypothesis, because it shows endemic species diverged long before the last Pleistocene glaciations (*DiBattista et al., 2015*). Although not

conclusive, the apparent stability of coral communities across the Middle and Late Pleistocene to the present suggests many coral species were able to persist in refugia, either inside or just outside the Red Sea. If species were required to recolonize the entire basin from the Indian Ocean, we might expect a higher turnover in species across glaciations, similar to what is seen in the molluscan fauna (*Taviani, 1998*).

Although not helpful in the short-term preservation of reefs, these findings give some idea about ensuring the long-term survival of reef systems. As recent work on the Great Barrier Reef makes clear (*Hughes et al., 2017*), global scale ocean warming poses the most significant threat to reef survival, independent of water quality, and fishing pressures, and such a threat cannot be dealt with through local governance. As Earth enters the third year of the most damaging and widespread bleaching event in recent history (*Coral Reef Watch (CRW), 2017*), perhaps conservationists should be looking for pockets of reef habitat where, for one reason or another, surface temperatures remain cool enough to prevent mass bleaching. These potential refugia should be legally protected to help ensure some coral populations survive into a theoretical future in which ocean warming has reversed and corals can repopulate former habitat.

## CONCLUSION

The three Middle Pleistocene terraces studied here preserve two distinct reef zones: two shallow back reefs and a shallow fore or back reef slope. The five Late Pleistocene terraces preserve a fore-reef slope, two depths on another fore-reef slope, a reef edge, and a reef flat. All of the Pleistocene reef zones have a species richness and community assemblage comparable to modern reefs of the Red Sea. Only one Middle Pleistocene species, *Favites micropentagonus*, has not been reported from the modern Red Sea, and may have gone locally extinct due to hypersaline conditions during a glacial high stand. Three other species, *Stylocoeniella guentheri*, *Astrea curta*, and *Pavona decussata*, are found in the Middle Pleistocene and modern Red Sea, but not in the Late Pleistocene. They may have also gone locally extinct during MIS 6, the glacial period following the Middle Pleistocene interglacial, and only recolonized the Red Sea at the onset of the current interglacial period. Modern Red Sea endemics are absent from the Red Sea Pleistocene record, although two species once believed to be endemic, *Erythrastrea flabellata* and *Favites vasta*, are reported for the first time from the fossil record, pointing to the Red Sea as a possible location for the origination of these species. The relative stability of coral communities in the Red Sea across multiple Pleistocene glaciation events suggests that if coral can survive in a few, small, protected areas, they can recolonize large areas given appropriate water conditions and time. The Pleistocene coral reefs of the Red Sea have a rich fossil record that spans nearly the entire coastline. Further sampling is needed to compile a taxonomic database that would allow for geographic, ecological, and temporal analyses to answer questions about endemism, global diversity, and biotic response to climate change.

## ACKNOWLEDGEMENTS

The author would like to thank two anonymous reviewers for their thorough and thoughtful feedback; their contributions have greatly improved the manuscript. The author

would also like to thank Haysam Odema and Timothy Pearson for logistical support and assistance in the field and Dr. Jere Lipps and Dr. Amin Strougo for helpful discussions. Special thanks to Drs. Diane Irwin and Mark Goodwin of the University of California Museum of Paleontology for assistance with thin sections and curation. Thanks are also due to the entire staff of the Binational Fulbright Commission in Egypt, especially Noha El Gindy, for her support in-country. The author also thanks Drs. Medhat Said Abdel Ghany, Abdel Latif, and Ehab El Saddy and their devoted staff at the Egyptian Geologic Museum in Cairo for their hard work and cooperation in processing fossil samples for shipment.

### Funding
Funding for this research was provided by a Fullbright Fellowship awarded by the U.S. Department of State, and by the Dorothy K. Palmer Fund, awarded by the University of California Museum of Paleontology. There was no additional external funding received for this study The funders had no role in study design, data collection and analysis, decision to publish, or preparation of the manuscript.

### Grant Disclosures
The following grant information was disclosed by the authors:
U.S. Department of State.
University of California Museum of Paleontology.

### Competing Interests
The author declares that they have no competing interests.

### Author Contributions
- Lorraine R. Casazza conceived and designed the experiments, performed the experiments, analyzed the data, contributed reagents/materials/analysis tools, wrote the paper, prepared figures and/or tables, reviewed drafts of the paper.

### Field Study Permissions
The following information was supplied relating to field study approvals (i.e., approving body and any reference numbers):

The field permit was issued by the Egyptian Environmental Assessment Agency (EEAA) and authorized by Dr. Moustafa Fouda. All fossil material was reviewed and approved for export by the Egyptian Geologic Museum in Cairo.

### Data Availability
The raw data has been supplied as Supplemental Dataset Files.

### Supplemental Information
Supplemental information for this article can be found online at http://dx.doi.org/10.7717/peerj.3504#supplemental-information.

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
