# Peer review of "Pleistocene reefs of the Egyptian Red Sea: environmental change and community persistence"

_PeerJ, doi:10.7717/peerj.3504_

## Round 0.1 · original submission · Major Revisions

Dear Dr. Casazza,

We have received the reports from two reviewers on your manuscript “Pleistocene reefs of the Egyptian Red Sea: environmental change and community persistence”. I must inform you that, based on the advice received, your manuscript may be considered acceptable for publication in PeerJ after a major review is performed. See below the detailed comments kindly provided by the two reviewers, which performed an extensive work. Please, consider all the referees remarks in your revised manuscript and indicate in your rebuttal letter a point-by-point response to the referees.

With kind regards,

Ronaldo

Reviewer 1 ·

Basic reporting

The manuscript is generally very well-written and appears to conform to all PeerJ policies related to the structure of the manuscript. There are a lot of single-sentence paragraphs throughout the manuscript. I would suggest combining these with the previous paragraph whenever possible. The data are made available in a supplemental file.

The figures are relevant and the captions provide sufficient description, but the order of the figures showing the fossil coral specimens seems a bit haphazard at present. I would organize so that they are grouped by either coral family or locality (or both). It’s also unclear why these particular images are shown. For example, in several of the figures there are multiple images showing the same taxa (often from the same locality). I would suggest showing single images of well-preserved “type-specimens” of the dominant genera only. Alternatively, if the authors have a compelling reason for including all the figures, it should be stated in the text of the manuscript. Additionally, I would strongly suggest that the coloration of coral genera be kept the same in Figures 19 and 20 so that readers can visually contrast the Middle and Late Pleistocene coral assemblages. Non-coral constituents should also be presented in a different pattern in those figures. Finally, if possible the author should include a figure similar to Figures 19 and 20 that shows the modern composition of the Egyptian Red Sea so that readers can evaluate the conclusion that the assemblages are similar.

Experimental design

The original, primary research described in the manuscript fall within the Scope of the journal. The author gives a good Introduction to the relevance of the study, but I suggest that she include an explicit statement about the objectives of the study at the end of the Introduction section. I also made some minor suggestions for the Methods section in the text of the manuscript, but the description and execution of the Methods are sufficient for this type of study.

Validity of the findings

The general conclusions of the study: that the coral assemblages of the Middle and Late Pleistocene are similar to those of the modern and, therefore, that glaciations did not drive significant changes in the composition of reefs in the region, seem valid; however, the author over-states or over-simplifies there conclusions in some parts of the manuscript. For example, there is no discussion of why modern endemics may be absent from the fossil record. This is a really interesting finding of the study that should be discussed because it has important implications for the origin of those taxa. There is also no formal discussion of the differences between the Middle and Late Pleistocene assemblages and what may explain those differences. The author should be careful about overly simplified statements like "Middle and Late Pleistocene taxa are the same as modern reef taxa".

I also don't think that the author presents enough information to evaluate their conclusions about the paleoenvironments in which the various Pleistocene coral communities grew. In the Results, each section describing one of the Pleistocene deposits begins with a broad statement about the type of reef environment, but the author does not give any supporting information to back up that conclusion. In the Discussion, the author uses the high abundance of Porites spp. as the main line of evidence that the Pleistocene reefs grew in turbid environments; however, Porites can also be abundant in low-sediment environments, so its high abundance alone does not prove that the reef was growing in a turbid environment. It would be good to discuss other indicators of high turbidity in more detail. The Methods suggests that sediment type and outcrop orientation were used as indicators of paleoenvironment, but this information is not included in the discussion or results.

Additional comments

This manuscript presents a new paleoecological reconstruction of the Pleistocene coral assemblages from an understudied but important region of the world: the Egyptian Red Sea. As the authors point out, the extreme environments of the Red Sea can provide insight into how coral reefs elsewhere may respond to climate change and other disturbances in the future. The authors’ unique dataset should, therefore, be of broad interest to coral-reef scientists. The manuscript is generally very well written; however, there are some sections (particularly in the Discussion) that would benefit from additional discussion and/or reorganization. Most of my most significant concerns have been outlined in the sections above, but I discuss a few additional points below and a number of more minor comments/suggestions are included as edits to the manuscript.

As mentioned in the previous section, my most significant suggestion is that the authors add a more detailed rationale for their conclusions about the Pleistocene paleoenvironments and that they add a discussion about why modern endemics were absent from the fossil record. Although the manuscript generally flows very well, I think that the author should also consider some reorganization of the Discussion section because it does not flow very well. One idea would be to organize the Discussion to address each of the specific objectives the author should add to the end of the Introduction. Additionally, the Taxonomy section at the beginning of the Discussion seems like it could either be moved into the results or combined with the “Community” sub-section. I would also consider including a more explicit discussion of the similarities and differences between Late and Middle Pleistocene assemblages.

Annotated reviews are not available for download in order to protect the identity of reviewers who chose to remain anonymous.

Reviewer 2 ·

Basic reporting

The article “Pleistocene reefs of the Egyptian Red Sea: environment change and community persistence” provides a dataset on the fossil reef terraces in the Red Sea collected 2007-2008. There is a high quality set of images on the fossil corals studied providing an excellent repository for coral taxonomy work.

However, coral taxonomy is not entire focus of the paper and unfortunately, due to the poor use of language with respect to expression, sentence and paragraph construction it is difficult to follow and grasp the science being communicated.

The introduction of the article requires the use of more up-to-date publications to provide a cohesive context to the study (for example, lines 42-44 on the threats to coral reefs worldwide and the use of one reference – Wilkinson, 2008). There are also a need for more explanation to support the study area background rather than referring to ‘some authors’ or ‘an exhaustive review’. Whilst there is some evidence of justification to the selected survey sites (specific terraces) there needs to be more rigour in pulling this together.

Experimental design

There is clear attempt to justify the field sites selected and eludes to some of the reasons why studying fossil reefs is relevant and meaningful today. More references needed for a stronger argument.

The methodology section of the article is very weak. A couple of examples:
Line 133: ‘record coral species on all examined terraces’
• Record what about coral species? Abundance? Size? Preservation?
Line 133-136: ‘A tape measure was laid along the terrace and photographs were taken at each meter mark to record the reef building coral species…. or sediment/substrate type present’
• Length of tape measure? units?
• How was is laid, direction relative to what?
• What type of camera? Aerial, oblique, scale?
• Whole coral colony? corallite detail?
• What categories were used for sediment/substrate type and why?

It is recommended that the use of ‘I’ in the methodology section not be overused; ‘I under point transects…’ (line 133), ‘I did transects…’ (line 142), ‘I selected field sites…’ (line 123), ‘I used transect data…’ (line 156).

Coral taxonomy: the author refers to the references used for coral species identification.
• Why the 'lowest taxonomic level' (line 150)? Indicate why this is.
• For the thin sections which methodology was this following and what equipment was used (for both preparation and analyses).
• Among other things it is stated that ‘…the presence/absence of turbid water species helped to determine the degree of hydrodynamic exposure.’ (lines 158-159) However, it is not clear whether you are referring to previously published studies which establish what species are considered ‘turbid water species’, what is the link between species and hydrodynamic exposure, and so forth.

The statistical analyses section of the methodology is vague:
• What program was used for analyses?
• Was data checked for normal distribution and to determine which were the most appropriate statistical tests to use?
• Is there not potential to use statistical analyses to determine whether there are any significant differences between the coral assemblages of each terrace and what species may be responsible for possible differences?
• The author indicates that their dataset is compared to modern Red Sea and regional datasets – what statistical techniques did this include or if none why, and how was this comparison carried out? Why are you comparing?

The study clearly states the relevant permits and regulations required to carry out the research.

Validity of the findings

The extensive set of images provides excellent reference material for the taxonomy of these fossil corals and for future coral taxonomy studies. The full dataset has been provided and is available for viewing.

The results section is systematic which can often be the case for multiple section/facies descriptions in order to allow direct comparison of specific features, however, it is also repetitive and oversimplified with continuous use of one sentence to form a paragraph, and/or the use of extremely abrupt ‘sentences’. For example, ‘Coral cover was 78%.’ (line 214, and the same in Line 200, 230, 244, 260-261, 274, 288, and 306). Need to make better reference and use of figures and data tables.

Line 165 the opening line of the results: ‘A total of 1003 data points were collected…’. There is no indication whatsoever of what these data points actually are.

In the discussion section the author needs to make it clear when the authors data is being discussed or when the author is referring to previously documented findings: e.g. lines 378-379, also see lines 360-362, it is not clear whether this is referring to fossil coral assemblages from the authors study, all terraces? They indicate they are consistent with modern reefs, but which modern reefs and documented by who?

Lines 370-377 talk to turbidity levels and the hydrodynamics of sediment removal – however, lacks sufficient referencing and there is no mention of sedimentation. What is the difference between sedimentation and turbidity? How does this relate to the different species assemblages that appear or the reef zones your interpret?

Unfinished sentences: e.g. lines 396-397: ‘Using cluster analysis they identified 13 distinct coral communities, some of which had subtypes.’ What is important about this? How does this link to your data?

Some parts of the discussion lack support from either data or previously published work (e.g. see lines 435-439).

There is attempt to mention life-death assemblages vs modern live coral measurements but it becomes a meaningless paragraph and not fully supported by the literature (lines 442-447).

Lines 448-450 are unsupported and do not form a paragraph.

Lines 484-488 are weak. It appears the author is alluding to the controls of coral growth and survival, or could be reef-building potential(?) - it is not clear. Either way there lacks a concise supported discussion on the difference between optimal vs marginal coral reef environments. See Perry & Larcombe (2003) for summary.

Lines 495-512 lack any reference to substrate availability and/or suitability and depth range for coral colonization during sea level rise. Or any clear statements on whether there are differences in coral diversity (and it relationship with the number of ‘less tolerant’ species) during different geological periods.

Overall the results and discussion section do not do justice to the dataset as it stands. It is recommended that the discussion points are reviewed and to ensure that they are clearly supported by the data and/or relevant literature.

Additional comments

The article “Pleistocene reefs of the Egyptian Red Sea: environment change and community persistence” provides a dataset and excellent photographic record of the fossil reef terraces in the Red Sea collected 2007-2008. However, this is not solely a taxonomy paper and unfortunately due to the poor use of language it is difficult to follow and grasp the science being communicated. Further, the methodology and discussion sections need significant work to ensure they are concise and supported. It is recommended that the author revisit the manuscript and get input from a coauthor or colleague to assist with this process.

---

## Round 0.2 · Minor Revisions

As you can see, the original reviewers are largely pleased with your revisions, although both recommend further minor changes. Please address all of their review comments in your revision.

Reviewer 1 ·

Basic reporting

The manuscript meets the standards of basic reporting.

Experimental design

The aims and broader context of the study is clear and there is generally enough information methods. My only major suggestion is that the author should include a description in the Methods section of how she determined whether the sediments encountered in the surveys were fine- or coarse-grained an how she determined percentage siliclastic vs. carbonate sediment. This data was used in the interpretation of some of the paleoenvironments, but there is no information about how the data was collected.

Validity of the findings

The revised manuscript provides a much clearer and well-organized discussion of the results. There are a few areas of the Discussion that I think warrant further discussion and/or clarification, which I highlight below and as edits to the text of the manuscript:

First, I would like to see the author include the information on the sediments (and/or coralline algae) encountered in the surveys in their interpretation of the paleoenvironments for the lower terrraces of the late Pleistocene. This information is included for the other sites, but it is missing form the Discussion of these sites.

Lines 586-590: There doesn't seem to be any consistent difference in diversity between the middle and late Pleistocene from the rarefaction curves. I would add a couple of sentences to discuss this.

Lines 592-593: I would flesh out a little more clearly why this is unexpected. Summarize what Pandolfi's studies found and how this result is different.

Concluding paragraph: I would add to this a bit so that the paper concludes with some broader implication of the study. You begin the manuscript by talking about modern coral-reef degradation. I would add a brief discussion of what the presence of historic refugia in this region say about the potential of reefs in the Red Sea to survive future anthropogenic perturbations.

Additional comments

This is an interesting study that provides new data for a critically understudied region. The revised manuscript is much improved and I am generally satisfied with the revised draft. It should be accepted for publication in Peer J after the minor revisions I have suggested.

A few other minor comments/suggestions are listed below and there are some more text edits suggested on the manuscript document:

Figures:
It would be good to include a map that shows the broader geographic context (with these places labeled) either as another panel to figure 1 or as a separate figure.

Figures 2-18: The author said that these figures were re-grouped by family, but there are still mixtures of families in the same figures. Most (Figures 3, 6-14, 16, 17) are ok, but multiple families are combined in the other figures. This may make sense for families where there are only a few specimens, but sometime the same genera (e.g., Galaxea) appear in multiple figures with other genera from different families. I would also suggest including the family names at the beginning of the caption and starting with the most common families at your sites and ending with the rare ones.

Figure 19: I would color the curves according to whether they are from the middle or late Pleistocene or, alternatively, by putative depth of their paleoenvironment.

There are also some typos, particularly in the results (words that shouldn’t be italicized, missing spaces, missing commas, etc.). I have highlighted many of these in the text, but I would suggest that the author do a careful copy edit of the manuscript.

Annotated reviews are not available for download in order to protect the identity of reviewers who chose to remain anonymous.

Reviewer 2 ·

Basic reporting

Despite large sections of the manuscript being rewritten there are still issues with the use of language for publishing purposes.

For examples:

Opening lines need editing.

"...while there is ongoing debate about whether reef taxa survived glacial periods within the Red Sea in hypothesized, lower salinity....reefs of the Egyptian coast were most likely devastated by glacial conditions."

"...making it a valuable resource for understanding coral response to a changing planet."

Methods section 'study area' needs to be abbreviated and tighter.

"..too eroded or damaged to identify even to the family level..."

"...These were all the sediment and substrate yes that were encountered and they provided useful information about the reef environment."

"...all of which correlate loosely with depth..."

lines 358-363
lines 397-399 - unclear whether your study or other study
lines 399-403 - unclear

heavy use of Chappell 1980 for discussion

from discussion: "...is within one standard deviation of this with 7 species found on a 110 m transect....likewise...is within one standard deviation of 9.37....."

"...The presence of fine-grained carbonate mud at Wadi Wizr further supports the idea that is was a low energy/sheltered environment, because very fine-grained sediments only accumulate in low energy environments." Who says? Where is supporting evidence for this?

Lines 543-550 are poorly written.

Despite major revisions there are also still some typos and a missing date through the manuscript. See lines 66, 139, 183, 199, 234, 239, 254, 463, 433, 476. Commas/colons: line 66, 67...

Experimental design

No comment

Validity of the findings

Where is conclusion?

Although there has been additions and restructuring in the discussion it is not entirely clear.

Second to last paragraph does not refer to any of your findings.

Additional comments

It is clear that many of the issues highlighted in the first review have been tackled, however, there are still some issues with the use of language and expression which I think need to be addressed for it to be distinguished from a chapter.

---

## Round 0.3 · Minor Revisions

I have been asked to step in as the previous Editor is unavailable.

It appears that the prior questions and concerns raised by the two reviewers have been adequately addressed. I understand that this manuscript has gone through a fairly lengthy review period and I will gladly work with you to see it to a speedy conclusion. Having looked through the manuscript myself I noticed a few items that I would like to see addressed before I can recommend acceptance for publication:

Underlying taxonomy:

You cite various works that you have used for identification of your samples and then state “Taxonomic names have been updated to reflect 182 revisions by Budd et al. (2012).” The Budd et al. paper dealt with the taxonomy of a single family, the Mussidae. How did you update/standardize the remaining groups? For instances, in the caption to Table 1, you state “Taxa were originally reported as a synonym of updated species name listed here” and then give the previous and a corrected name. I noticed that many of your names are not reflecting current taxonomic treatments. For instance, Goniastrea australense is now considered Paragoniastrea australensis (see Huang, D.; Benzoni, F.; Arrigoni, R.; Baird, A. H.; Berumen, M. L.; Bouwmeester, J.; Chou, L. M.; Fukami, H.; Licuanan, W. Y.; Lovell, E. R.; Meier, R.; Todd, P. A.; Budd, A. F. (2014). Towards a phylogenetic classification of reef corals: the Indo-Pacific genera Merulina , Goniastrea and Scapophyllia (Scleractinia, Merulinidae). Zoologica Scripta. 43: 531-548., available online at https://doi.org/10.1111/zsc.12061 ). Other names in the manuscript are misspelled (e.g., Canthareullus should be Cantharellus, Paramontastrea should be Paramontastraea). I strongly suggest comparing your nomenclature against the entries in WoRMS (www.marinespecies.org).

Specimen vouchering:

You state: “Specimens of the most commonly occurring and difficult to identify reef species were collected and deposited at the University of California Museum of 184 Paleontology (coral specimen numbers 557184 - 557253, other invertebrate specimen numbers 185 557273 - 557391).“

This is rather vague and does not really help with future re-investigations of the material. Please provide individual collection voucher numbers with each figure caption.

PeerJ conventions:

Please standardize the scale bars in your figures to a single style.

Reviewer 1 ·

Basic reporting

The manuscript is in good shape. I noted a (very) few typos and formatting error on the document.

Experimental design

The methods clear and appropriate for the study.

Validity of the findings

The discussion of the results is thorough and sound.

Additional comments

I made some minor editorial suggestions, but I suggest that the manuscript be accepted for publication.

Annotated reviews are not available for download in order to protect the identity of reviewers who chose to remain anonymous.

---

## Round 0.4 · Minor Revisions

This looks very good. Just a couple of minor technicalities that need addressing:

(1) The explanation of name changes in the caption to Table1_REV.docx can be misinterpreted:

For instance, your statement “Astrea curta originally reported as Montastrea curta (Hoeksema 2014)” can be understood as Hoeksema having it originally reported as Montastrea curta, or as Hoeksema (2014) being the taxonomic authority for the taxon Montastrea curta.

Both interpretations would be wrong – what you mean to say, I believe, is “Astrea curta originally reported as Montastrea curta (new placement following Hoeksema, 2014).”

Please adjust this throughout.

(2) For SpecimenCollectionUCMP.xlsx, you seem to have checked the coral names against the WoRMS nomenclature, but not the other taxa (e.g., the current name for the gastropod Cymatium aquatile is Monoplex aquatilis). Please update these as well (unless you disagree with the taxonomic decisions). In these not-previously cited (non-coral) cases, there will be no need to reference the WoRMS sources.

---

## Round 0.5 · accepted · Accept

Thank you for your revised manuscript. I am satisfied with your response to the referee comments, and am now happy to move this forward into production.